# Polyhedral Embeddings and Realizations of Orientable and Non-Orientable Cubical Surfaces using Reinforcement Learning

## Abstract

Finding realizations in $\mathbb{R}^3$ of polyhedral maps on compact connected surfaces is considered a hard problem in discrete geometry because of the lack of general solution methods. Heuristic approaches have been proven efficient in finding polyhedral embeddings of orientable vertex-minimal surfaces of genus $g$ by minimizing their intersection length; however, they can still be challenging to implement due to large configuration spaces, and can struggle avoiding local minima.

This article studies closed connected cubical surfaces; surfaces made from a collection of faces of a 5-dimensional cube. The author proposes a Reinforcement Learning (RL) algorithm to minimize the number of face intersections of orientable and non-orientable cubical surfaces through 5-dimensional rotations or modifications on the perspective projection distances; yielding immersions that are perspective projections of a unitary 5-dimensional cube. Polyhedral embeddings of orientable cubical surfaces of genus $g = 1, 2$ and realizations of the Projective Plane and the Klein Bottle with the smallest possible number of face intersections are obtained. The agent's optimal strategy is visualized using three-dimensional animations.

## 1 Introduction

### 1.1 Polyhedral Realizations by Triangles

Compact Surfaces are 2-dimensional topological manifolds and are completely classified. There are two infinite families of compact surfaces, the **orientable surfaces** of genus $g \geq 1$, which are torus with $g$ "handles", and the **non-orientable surfaces** of demigenus $k \geq 1$. There is also the sphere which is orientable and has no "handles". Topological manifolds are often too general to work with directly; for this reason, it is often essential to assume the **triangulability of manifolds**. The problem of finding a minimal triangulation of manifolds has been widely studied and has been completely solved for compact surfaces. In the context of simplicial surfaces, a **minimal triangulation** can be defined as a triangulation using the minimal number of vertices, edges, and faces. It can be shown using Euler characteristic that the minimal triangulation in each of these three meanings of minimality is realized by the same triangulation up to isomorphism.

A **polyhedral map** on a surface is a finite set of polygons with at least three sides (usually triangulations) such that the intersection of any two distinct faces is either empty, a common vertex, or a common edge. Given a polyhedral map, it is natural to try to visualize it in $\mathbb{R}^3$ as a three-dimensional polyhedron or as a projection of a polytope in $\mathbb{R}^n$ such that every polygon is the convex hull of its vertices and two polygons are either: disjoint in $\mathbb{R}^n$, they intersect at a common edge and are not coplanar, or they intersect at a common vertex. This polyhedron is called a **polyhedral realization**.

A (topological) **embedding** is a continuous mapping that is a homeomorphism onto its image, while an **immersion** is a continuous mapping that is locally a homeomorphism. The image of an embedding doesn't have **self-intersections**, while the image of an immersion may have. From algebraic topology we know

that all orientable surfaces can be embedded in $\mathbb{R}^3$, while non-orientable surfaces can be embedded in $\mathbb{R}^4$ but only immersed in $\mathbb{R}^3$; however, for an orientable polyhedral surface, its embeddability depends on its defining triangulation; certain triangulations of orientable surfaces (usually minimal triangulations) are not realizable in $\mathbb{R}^3$ as shown by Bokowski & Guedes de Oliveira (2000). Consequently, in $\mathbb{R}^3$, any polyhedral realization of a non-orientable surface will have self-intersections, while polyhedral realizations of orientable surfaces without self-intersections are in some cases possible.

Hougardy et al. (2006) found polyhedral embeddings of surfaces of genus $g = 3, 4$ and examples of polyhedral realizations of genus $g = 5$ with 12 vertices. Their approach minimizes an intersection segment functional, which is the sum of the lengths of all face intersection lines occurring in the realization. Their technique can be described as a local search process, meaning that the final embedding is at most $T$ steps away from the initial immersion. For a given triangulation, the algorithm consists of assigning its vertices random integer coordinates in $\mathbb{R}^3$, ensuring they are in general position to avoid degenerate triangle intersections. Then, a vertex $v \in \mathbb{R}^3$ is randomly selected as well as a coordinate direction $\pm X, \pm Y$ or $\pm Z$ in which $v$ is moved an integer step. If the resulting set of coordinates is in general position and the new value of the intersection segment functional is strictly smaller than before, the movement is accepted and the next step is executed; otherwise, the movement is discarded and the process is repeated with the previous set of coordinates. The algorithm ends if the intersection segment length drops to zero, which means that a realization has been found. If after $T$ steps no realization has been found, then the vertices are assigned new initial random coordinates in $\mathbb{R}^3$ and the algorithm resets.

Brehm & Leopold (2016) extend this algorithm to find **realizations of non-orientable surfaces**; that is immersions with flat full-dimensional faces. Their improvement consists of modifying the objective function to minimize the edge intersection length of the faces contradicting an immersion, precisely the intersections of faces adjacent to a vertex. Such a vertex is called a **pinch-point**; a kind of surface singularity that violates the definition of an immersion. With this methodology and imposing suitable symmetry conditions that reduce the number of parameters and speed up the search, they found numerous new realizations of non-orientable surfaces with the minimal (or few) number of vertices. Some of them include the Projective Plane with one or two handles and the Klein Bottle with one or two handles. Before this improvement, the only vertex-minimal realizations of non-orientable surfaces were obtained by explicit construction; Brehm (1990) constructed examples for the projective plane with 9 vertices, and Cervone (2001) found examples of the Klein Bottle using 9 vertices; both proving that realizations with fewer vertices can't be constructed.

It is natural to ask whether it is always convenient to follow a greedy strategy or if, for certain surfaces and initial vertex coordinates, a solution can not be found without needing to increase the intersection length functional at some step. Here is where the idea of using **Reinforcement Learning** (RL) comes into play, because this technique can capture whether, in the long run, it is worth increasing the intersection segment to find a better realization or an embedding.

## 1.2 Polyhedral Realizations by Quadrilaterals

In their article on polyhedral surfaces of high genus, Ziegler (2008) study in particular **cubical surfaces**; two-dimensional **cubical complexes** of the $n$-dimensional cube homeomorphic to compact surfaces.

Govc (2024) gives a complete classification for $n = 6$ in terms of their genus $g$ for orientable cubical surfaces and their demigenus $k$ for the non-orientable. Non-orientable cubical surfaces first appear for the cube of dimension $n = 5$, however they appear also as surfaces in the 6-cube with one coordinate fixed. As with triangulations of closed surfaces, one can ask what the minimal cubical embedding of a given cubical surface is. Still, here minimality has to be specified more carefully since minimizing a cubical embedding can refer to vertices, edges, faces, or the dimension of the cube. In the following, a **minimal cubical surface** will refer to a surface that can not be realized with less faces from the $n$-dimensional cube $Q^n$.

**Quadrilateral realizations** in $\mathbb{R}^3$ of orientable and non-orientable 5-dimensional cubical surfaces are not yet studied. In this article, following the ideas in Hougardy et al. (2006), the author proposes a first approach using RL to realize perspective projections of 5-dimensional cubical surfaces in $\mathbb{R}^3$ with the **smallest number of face intersections** in $\mathbb{R}^3$, that is, the smallest number of pairwise intersecting quadrilaterals in $\mathbb{R}^3$. Polyhedral embeddings for orientable cubical surfaces of genus $g = 1, 2$ are obtained. Since the objective

is to minimize face intersections, pinch-points are allowed; such a realization is instead called a **singular realization**. Singular realizations are obtained for orientable cubical surfaces with $g = 3, 4, 5$ and non-orientable surfaces with demi-genus $k = 1, 2, 3$. In particular, the Projective Plane ($k = 1$) and the Klein Bottle ($k = 2$) here obtained can not be realized with less face intersections. They can be thought of as quadrilateral models of the **cross-cap disk** and the **pinched torus Klein Bottle** respectively, coming from perspective projections of a unitary 5-dimensional cube.

The realizations are obtained by projecting a realization on the 5-dimensional unitary cube to $\mathbb{R}^3$ via successive **perspective projection** maps. We assume that for a given initial **rotational orientation** of the cubical surface around the origin in $\mathbb{R}^5$, the **n-dimensional camera** is set at a position $c_5 \in \mathbb{R}^5$; from which projection rays extend to the 5-dimensional realization onto a **projection hyperplane** $p_5 \in \mathbb{R}^5$. The result of this first projection is a realization in $\mathbb{R}^4$, therefore, we can repeat the same process with a camera $c_4 \in \mathbb{R}^4$ and a hyperplane $p_4 \in \mathbb{R}^4$ to obtain a realization in $\mathbb{R}^3$.

The initial projection in $\mathbb{R}^3$ has an initial number of face intersections. The RL agent's actions consist of applying a 5-**dimensional rotation** or changing the **camera positions** ($c_5$ and $c_4$) in the perspective projections, sequentially modifying the number of face intersections. Since the realizations here obtained are always perspective projections of faces of a unitary 5-cube, vertices of the unitary 5-cube do not move individually and in integer steps, like in Hougardy et al. (2006) and Brehm & Leopold (2016); instead, they move according to a 5-dimensional $\epsilon$-degree rotation around some rotation plane in $\mathbb{R}^5$. However, this restriction on the movement of the vertices yields polyhedral realizations with notable symmetries, although deformed by perspective projection.

### 1.3 Main Contributions

The main contributions of this study are the following:

1. **Reinforcement Learning algorithm to minimize face intersections of a quadrilateral realization.** The author proposes an RL approach to find realizations (allowing pinch-points) of 5-dimensional orientable and non-orientable closed cubical surfaces with the minimum number of face intersections.

2. **Singular quadrilateral realizations** of the minimal cubical Projective Plane and the cubical Klein Bottle with the minimum number of face intersections achievable (3) and **quadrilateral embeddings** of minimal orientable cubical surfaces of genus $g = 1, 2$ are found.

3. **Animations of multiple face-minimization sequences.** For each cubical surface $\mathcal{C}$, the trained models return a sequence of steps consisting either of a camera modification or a 5-dimensional rotation, which can be used to build animations. The optimal strategy has the property that it allows a face intersection increase if necessary to find a realization. The author gives a link to the animation sequences in Section 8.

## 2 Background

### 2.1 Cubical Complexes and Surfaces

Cubical complexes have their origin in the beginning of the 20th century, with the work of Henri Poincaré and Solomon Lefschetz. The main definitions on cubical complexes and cubical homology can be consulted in Kaczynski et al. (2004), they serve as the mathematical foundations to define a cubical surface; a type of closed polyhedral surface made out from faces of an $n$-dimensional cube studied by Ziegler (2008). To define formally a cubical surface, denote the $n$-dimensional unit cube by $Q^n = [0,1]^n = [0,1] \times \cdots \times [0,1]$ ($n$ times), and its set of vertices by $Q_0^n$. Each vertex $v \in Q^n$ can be represented by an element of the set of all $n$-tuples with binary entries $\{0,1\}^n$, for example, the vertices of the unit square are represented by the set of tuples $\{(0,0),(0,1),(1,0),(1,1)\} \in \mathbb{R}^2$. The one-skeleton $Q_1^n$ is a graph with

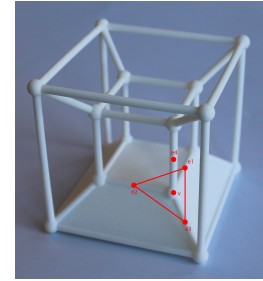

**Figure 1:** $\mathcal{F}_v$ on a 4-d Cubical Surface homeomorphic to a sphere.

vertex set $Q_0^n$ with an edge $e \in Q_1^n$ between two vertices if and only if they differ in exactly one coordinate. Regarded as a set, $Q_1^n$ consists of the set of vertices $v$ and edges $e$ of $Q^n$. The two-dimensional skeleton of $Q^n$ is denoted by $Q_2^n$ and consists of the set of vertices $Q_0^n$, the one-dimensional skeleton $Q_1^n$, and all its two-dimensional faces $f \in Q^n$. This construction can be continued up to the $n$-cube itself $Q_n^n$, and the elements of all the preceding sets are called the cells of $Q^n$. Every cell of $Q^n$ is a product of vertices and intervals, and therefore can be represented combinatorially as an element of $\{0, 1, 2\}^n$, where a 2 in an entry implies that in the product, the whole unit interval is considered. Thus, every sub-complex of $Q^n$ can be represented as a subset of $\{0, 1, 2\}^n$. A subset of $Q_2^n$ is called a **two-dimensional cubical complex** which in the following is denoted by $\mathcal{C}$, with sets of vertices, edges, and faces denoted by $\mathcal{C}_0$, $\mathcal{C}_1$, and $\mathcal{C}_2$ respectively. The **vertex figure** $\mathcal{F}_v$ of a vertex $v \in \mathcal{C}_0$ is the graph whose nodes are the edges in $\mathcal{C}_1$ having $v$ as an endpoint and where two nodes $e, e' \in \mathcal{C}_1$ are joined by an edge if there is a face $f \in \mathcal{C}_2$ with $e, e'$ as two of its edges. A **closed cubical surface** is a two-dimensional cubical complex $\mathcal{C}$ in which every point has an open neighborhood homeomorphic to an open disk. This condition is equivalent to requiring the following two conditions on $\mathcal{C}$: (1) Every edge is shared by exactly two faces, i.e., for all $e \in \mathcal{C}_1$, $F_e = 2$. (2) The vertex figure $\mathcal{F}_v$ of any vertex $v \in \mathcal{C}_0$ is a cyclic graph.

For dimension $n = 4$ only orientable cubical surfaces can exist, and the quadrilateral realizations in $\mathbb{R}^3$ of each cubical surface representative can be consulted in Estévez et al. (2023). In Section 5, realizations (some of them with pinch-points) of the perspective projections of minimal 5-dimensional cubical surfaces for genus $g$ $(1 \le g \le 5)$ and demigenus $k$ $(1 \le k \le 3)$ are presented. Non-orientable surfaces with $k = 1$ and $k = 2$ are equivalent to the Projective Plane and the Klein Bottle.

## 2.2 Face-Intersection minimization Process

In this work, given a 5-dimensional cubical surface $\mathcal{C}$ projected to $\mathbb{R}^3$ by perspective projection, the author finds quadrilateral realizations (allowing pinch-points) that minimize the face intersection number; that is the number of pairs of faces that have a non-empty intersection. This is a slightly different approach from the one from Hougardy et al. (2006), where the intersection segment functional is the quantity to minimize. However, instead of minimizing the face intersection number by moving the vertices $v \in \mathcal{C}$ individually, here they move according to the following kinds of linear transformations.

Rotations are an example of an isometry, they preserves Euclidean distances, so after applying a unitary 5-cube some rotation, the result is a unitary 5-cube. Consider two canonical vectors $\boldsymbol{e}^{(i)}, \boldsymbol{e}^{(j)} \in \mathbb{R}^5$ and let $X_{i,j} \subset \mathbb{R}^n$ be their span. In $\mathbb{R}^n$ there are $\binom{n}{2} = n(n-1)/2$ possible pairings $\boldsymbol{e}^{(i)}, \boldsymbol{e}^{(j)} \in \mathbb{R}^5$ or equivalently possible rotation planes $X_{i,j}$ yielding 10 possible $X_{i,j}$ for $n = 5$. A 5-dimensional rotation by an angle $\phi \in [0, 2\pi)$ fixing the plane $X_{i,j}$ is called a **5-dimensional elemental rotation** and can be represented by an elemental rotation matrix $\boldsymbol{R}_{i,j}(\phi) \in SO(5)$; elemental rotations generate all possible rotations in $\mathbb{R}^5$, and some aspects like their non-commutativity or Gimbal-Lock are discussed in Appendix A.3.

**Perspective projection** is used to visualize $n$-dimensional objects in $\mathbb{R}^3$ preserving linear segments from the $n$-dimensional object; it projects edges (line segments) $e \in Q_1^n$ into line segments $L \subset \mathbb{R}^3$ and faces (plane segments) $f \in \mathcal{C}_2$ into plane segments $T \subset \mathbb{R}^3$. It is a natural choice when intersections from projected line or plane segments in $\mathbb{R}^3$ must be computed. From a **camera position** $\boldsymbol{c}_n \in \mathbb{R}^n$, it maps a vector $\boldsymbol{a} \in \mathbb{R}^n \setminus \{\boldsymbol{c}\}$ to an orthogonal hyperplane $\boldsymbol{p} \in \mathbb{R}^n$, returning a projected vector $\boldsymbol{b} \in \mathbb{R}^{n-1}$. Perspective projection is then denoted by $Pr_n(\boldsymbol{a}_n, \boldsymbol{c}_n, \boldsymbol{e}_n) : \mathbb{R}^n \to \mathbb{R}^{n-1}$, mapping $(\boldsymbol{a}_n, \boldsymbol{c}_n, \boldsymbol{e}_n) \mapsto \boldsymbol{b}$ and its associated linear transformation is calculated as in Algorithm 2. Note that projected vertices $v \in \mathcal{C}$ move when affecting the camera position $\boldsymbol{c}_5 \in \mathbb{R}^5$ (resp. $\boldsymbol{c}_4 \in \mathbb{R}^4$). A further discussion can be found in Appendix A.4.

After rotating $\mathcal{C}$ by a "small" $\epsilon$-degree rotation $\boldsymbol{R}_{i,j}(\epsilon)$ or affecting the camera positions $\boldsymbol{c}_5 \pm \delta \boldsymbol{e}^{(5)}$ or $\boldsymbol{c}_4 \pm \delta \boldsymbol{e}^{(4)}$ by a "small" distance $\delta$, the projection of $\mathcal{C}$ has a new orientation encoded by a vector $s_t \in \mathbb{R}^{27}$ (see Section 3) and some **face intersection number** $FaceInt_{s_t}$ parametrized by $s_t$. The task is to find a sequence of transformations leading to a final state $s_T$ where $FaceInt_{s_T} \le Exp$, where $Exp \in \mathbb{Z}^+$ is the **expected minimum number of face intersections**. Moreover, the realization must have non-overlapping edges $Q_1^5$ (see Section A.2). Figure 2 shows an initial realization of cubical Projective Plane with $FaceInt_{s_t} = 17$ being transformed into a realization with $FaceInt_{s_T} = 3$ shown in Figure 26 which is the smallest number of face intersections possible for this cubical surface. This discussion will be formalized in Section 4.

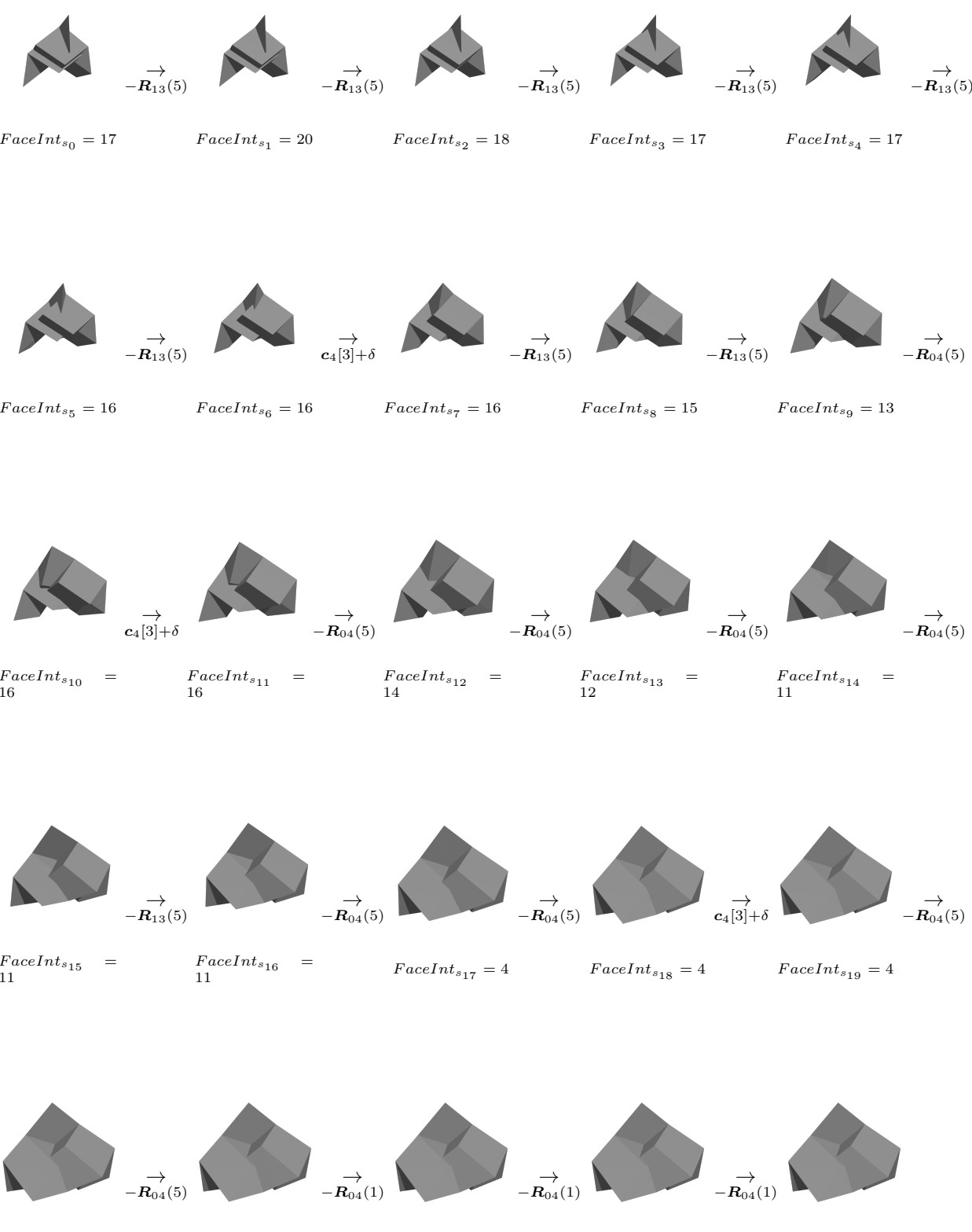

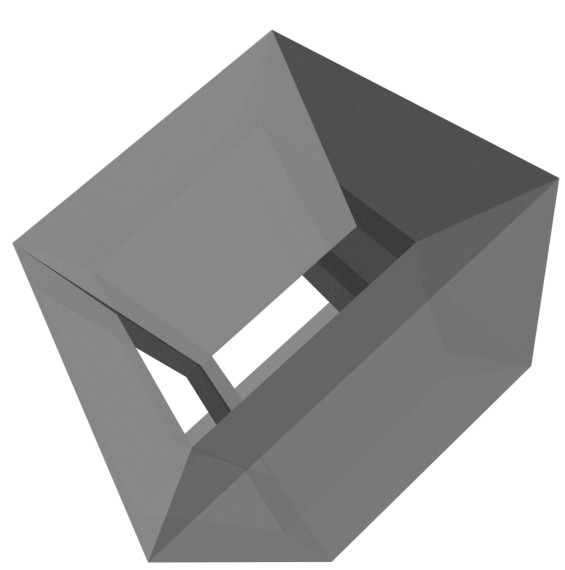 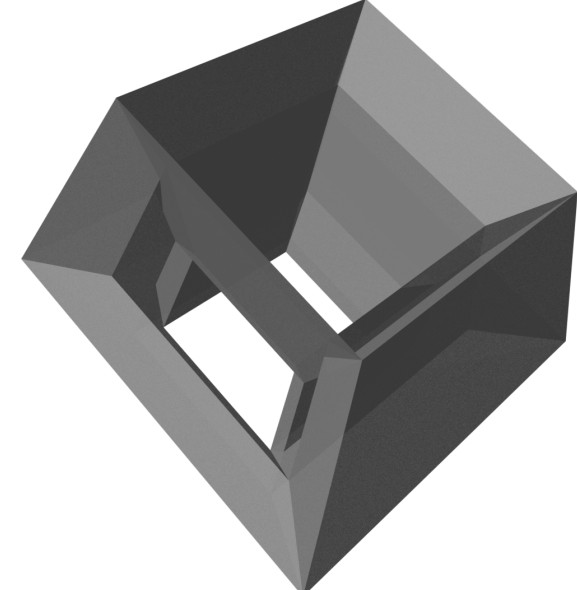

**Figure 27:** Cubical surface with $g = 5$ and $FaceInt_{s_T} = 16$ (side view).

**Figure 28:** Cubical surface with $g = 5$ and $FaceInt_{s_T} = 16$ (top view).

## 2.3 Orientable Cubical Surfaces

Orientability of a polyhedral surface does not imply its realizability in $\mathbb{R}^3$, the latter depends also on the number of faces building it. An orientable cubical surface with $g = 1$ can be realized in the 4-cube as shown in Estévez et al. (2023), so it can also be realized in the 5-cube as in Figure 47. This serves as a first test for the algorithm here presented. On the other hand, a cubical surface with $g = 2$ can only be constructed on cubes of dimension $n \geq 5$, and the RL algorithm successfully finds a polyhedral embedding in $\mathbb{R}^3$ shown in Figure 48.

Figures 27 and 28 show a singular realization of a cubical surface with $g = 5$. This is a (double) perspective projection of a unitary 5-cube, and despite being a singular realization it has some similarities with the "deformed" realization of a $g = 5$ cubical surface built by Ziegler (2008), where "deformed" means that the realization does not come from a perspective projection of a unitary 5-dimensional cube since some vertices are allowed to move independently. The deformation induced by perspective projection also plays a role in impeding some faces from intersecting. This was already noted by Hammack (2024) who builds a realization of a $g = 5$ cubical surface by introducing an "informal perspective" in which instead of calculating perspective projection with one camera point, it is allowed to "split" into two camera points.

## 2.4 Non-Orientable Cubical Surfaces

### 2.4.1 The Real Projective Plane

The real Projective Plane is a closed non-orientable surface that classifies all the lines in $\mathbb{R}^3$ passing through the origin. There are multiple singular and non-singular immersions of this surface in $\mathbb{R}^3$ like the **Boy's surface** or the **Cross-cap disk model** which is a singular immersion because it contains two pinch-points. However, the cross-cap disk model has a relevant property for this work; it is the singular realization that minimizes the intersection-line components having just one component. It can be built by gluing the boundary of a cross-cap with a sphere with a disk removed as explained by Francis & Weeks (1999). One can expect that the singular realizations of the cubical projective plane minimizing face intersections are quadrilateral realizations of a cross-cap disk model, and this is the case of the realization found by the RL algorithm in Figure 32. This singular realization is built gluing a quadrilateral cross-cap shown in Figures 29 and 30 with a quadrilateral "disk" shown in Figure 31. Note that in this singular realization there exist only one

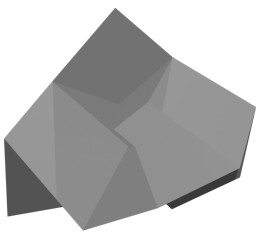
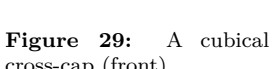

**Figure 29:** A cubical cross-cap (front).

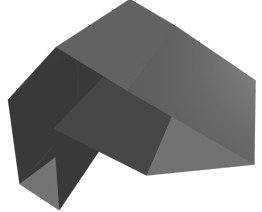

**Figure 30:** A cubical cross-cap (back).

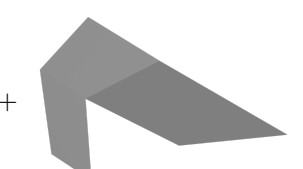

**Figure 31:** Three faces homeomorphic to a disk.

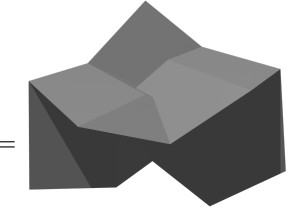

**Figure 32:** A cubical cross-cap disk model of the Projective Plane.

intersection-line component composed of 3 line segments, that is $FaceInt_{s_T} = 3$, and by the discussion in Appendix A.1 this is the lowest number of face intersections that can be obtained for any realization of the cubical Projective Plane.

### 2.4.2 The Klein Bottle

The **Pinched-Torus Klein bottle** is a singular realization of a Klein Bottle in $\mathbb{R}^3$ which can be thought of as a genus-1 torus whose boundary is flattened, allowing its external side to connect with the internal side through the intersection-line. Another interesting construction consists on joining two Möbius Strips along their boundaries. Figure 33 shows a cubical Möbius Strip made up from 6 faces of the 5-cube. It is possible to find a Möbius Strip in the 4-cube as shown in Estévez et al. (2023), however the Klein Bottle can only be built in cubes of dimension $n \geq 5$. Figure 35 shows the two disjoint Möbius strips in Figure 34 joined along their boundaries by the faces in gray. Note that in this singular realization there exist only one intersection-line component composed of 3 line segments, that is $FaceInt_{s_T} = 3$, and by the discussion in Appendix A.1 this is the lowest number of face intersections that can be obtained for any realization of the cubical Klein Bottle.

## 3 Reinforcement Learning

**Reinforcement Learning** (RL) is a machine learning technique proposed by Sutton et al. (2018). In RL, an algorithm called an agent interacts with its environment $\mathcal{E}$ by performing a sequence of actions maximizing a cumulative reward based on feedback received for each action taken. More specifically, at each time-step $t$ the agent takes as input information from the environment called a **state** $s_t \in \mathcal{S}$ (this is the information that the agent knows about $\mathcal{E}$) and outputs an **action** $a_t \in \mathcal{A}$ which is then passed to $\mathcal{E}$; returning a new state $s_{t+1} \in \mathcal{S}$ and a **reward** $r_t$ for taking $a_t$ at $s_t$. Future rewards are multiplied by a **discount factor** $\gamma \in [0, 1)$ at each step. The **expected discounted return** at step $t$ is defined as $R_t = \sum_{t'=t}^{T} \gamma^{t'-t} r_t$. A **policy** $\pi : \mathcal{S} \to \mathcal{P}(\mathcal{A})$ is a map from the states to the set of probability distributions over actions mapping

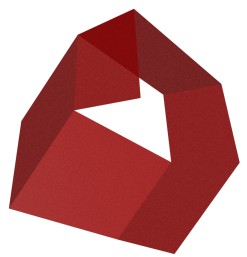

**Figure 33:** A 6-faced cubical Moebius Strip.

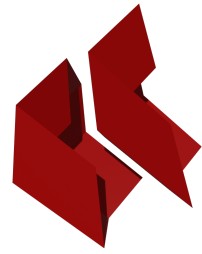

**Figure 34:** Two disjoint cubical Moebius Strips.

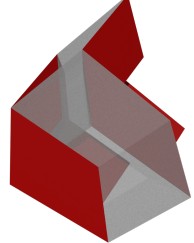

**Figure 35:** Both copies join to build a cubical pinched-torus Klein Bottle.

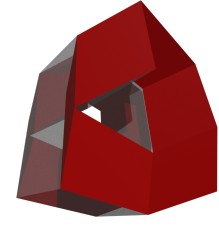

**Figure 36:** Cubical pinched-torus Klein Bottle.

$s \mapsto \pi(a|s)$, where $\pi(a|s)$ is the conditional probability of selecting the action $a$ at the state $s$. The **state-value function** following a policy $\pi$ is then defined as $v_\pi(s) = \mathbb{E}_\pi \left[ r_t + \gamma r_{t+1} + \gamma^2 r_{t+2} + ... | s_t = s \right]$; that is the expected value of $R_t$ by following $\pi$. The **action-value function** following a policy $\pi$ defined as

$$Q^\pi(s, a) = \mathbb{E}_\pi \left[ R_t | s_t = s, a_t = a, \pi \right], a_t \sim \pi(\cdot|s_t) \tag{1}$$

is the expected value of $R_t$ by taking an action $a \in \mathcal{A}$ and following $\pi$ afterwards. Let $\Pi$ be the set of all policies, the **optimal action-value function** is the maximum expected value of $R_t$ achievable by following any $\pi \in \Pi$ after performing $a \in \mathcal{A}$ at $s \in \mathcal{S}$; that is $Q^*(s, a) = max_{\pi \in \Pi}\{\mathbb{E}_\pi \left[ R_t | s_t = s, a_t = a, \pi \right]\}, a_t \sim \pi(\cdot|s_t)$.

### 3.1 Proximal Policy Optimization

Proximal Policy Optimization Algorithms (PPO) are a family of RL algorithms that compute an estimation of the policy gradient and plug it into a stochastic gradient ascent algorithm. Among this family of algorithms, we use the Clipped Surrogate Objective algorithm proposed by Schulman et al. (2017), which attempts to maximize the objective function $L^{CLIP}(\theta)$ with actor network weights $\theta$ and critic network weights $\varphi$. The actor network takes as input a state $s_t$ and outputs an action $a_{t+1}$, while the critic network takes as input a state $s_t$ and outputs the value $V_\varphi(s_t)$ of the state $s_t$. Let $V_\varphi(s_t)$ be the this state-value function $v_\pi(s)$, and $r_t = \pi_\theta(a_t|s_t)/\pi_{\theta_{old}}(a_t|s_t)$ the probability ratio, the **advantages** at state $s_t$ are calculated as

$$\hat{A}_t = Q^{\pi_{\theta_{old}}}(s_t, a_t) - V_{\theta_{old}}(s_t), \tag{2}$$

and the clip objective function to maximize is

$$L^{CLIP}(\theta) = \mathbb{E}_t[min(r_t(\theta)\hat{A}_t, clip(r_t(\theta), 1 - \epsilon, 1 + \epsilon)\hat{A}_t)], \tag{3}$$

where in common practice $\epsilon = 0.2$. The actor and critic network weights are updated independently by stochastic gradient descent; in this case ADAM optimizer by Kingma & Ba (2017).

## 4 Minimizing Self-Intersections of Cubical Surface Realizations

A Markov Decision Process (MDP) is a tuple $M = (\mathcal{S}, \mathcal{A}, \mathcal{R}, \mathcal{P}, \gamma)$ where $\mathcal{S}$ is a state space, $\mathcal{A}$ an action space, $\mathcal{P}$ a transition probability function, $\mathcal{R}$ a reward function and $\gamma \in [0, 1)$ a future reward discount factor. Solving $M$ means finding a policy $\pi$ over $\mathcal{A}$ yielding the supremum of $R_t$, that is finding $\pi$ maximizing at each state the state-value function $v_\pi(s)$.

Finding quadrilateral realizations of $n$-dimensional cubical surfaces minimizing face intersections can be formulated as an MDP. For the rest of this article, fix $n = 5$ and consider a 5-dimensional cubical surface $\mathcal{C}$. The quadrilateral realization of a cubical surface $\mathcal{C}$ in $\mathbb{R}^3$ is expected to have a positive number of face intersections, especially if $\mathcal{C}$ is non-orientable. To minimize them, at each state $s_t \in \mathcal{S}$, the **face intersection number** $FaceInt_{s_t} \in \mathbb{Z}^+$ is calculated as in Algorithm 14. It determines whether the projection of two faces $f_1, f_2 \in \mathcal{C}$ intersect by dividing the projection of each face into two triangles $\{T_1^i, T_2^i\}$ for each $f_i$ and using triangle collision detection for each possible pairing of triangles $T_i^1, T_j^2$. Appendix A.9 describes the Algorithm by Möller (1997) determining whether two triangles $T_1, T_2 \subset \mathbb{R}^3$ intersect and the coordinates of their intersection line or point. A Python implementation of the algorithm by NeonRice (2020) is used in this work. In general, for $\mathcal{C}$ orientable or non-orientable, it is unknown how few face intersections one can achieve; however an **expected minimum** face intersection number $Exp \in \mathbb{Z}^+$ can be set as an objective. At $s_t$ the agent attempts to modify $FaceInt_{s_t}$ with respect to the previous number of face intersections $FaceInt_{s_{t-1}}$ from two different approaches, depending on the **exact parameter** $Exact$. If the parameter $Exact = True$, the agent attempts to strictly set $FaceInt_{s_t} = Exp$, otherwise it is enough to set $FaceInt_{s_t} \leq Exp$. If some of these conditions is achieved, the task enters a final stage. Consider the edges $e \in Q_1^5$ of the 1-skeleton. After projecting each edge, the resulting line segments are assigned a width $w > 0 \in \mathbb{R}$ and two projected edges $L_{1,2}, L_{3,4} \subset \mathbb{R}^3$ overlap if the shortest line segment connecting them $L_{a,b} \subset \mathbb{R}^3$ has length $|L_{a,b}| \leq w$. A formal definition of the **number of edge overlaps** $Overlap_{s_t} \in \mathbb{Z}^+$ at state $s_t$ is presented in Appendix A.2. The agent attempts to find a state $s_t$ at which $Overlap_{s_t} = 0$ and $(FaceInt_{s_t} = Exp$ or $FaceInt_{s_t} \leq Exp)$.

### 4.1 State Space

The parameters needed to build the realization of a 5-dimensional cubical surface $\mathcal{C}$ in $\mathbb{R}^3$ are of two kinds. The 5-dimensional (resp. 4-dimensional) **camera distance** $d_5 \in [-15, -2] \in \mathbb{R}$ (resp. $d_4 \in [-15, -2] \in \mathbb{R}$) is a scalar value representing the 5-dimensional (resp. 4-dimensional) camera position $\boldsymbol{c}_5 = (0, 0, 0, 0, d_5) \in \mathbb{R}^5$ (resp. $\boldsymbol{c}_4 = (0, 0, 0, d_4) \in \mathbb{R}^4$). The surface $\mathcal{C}$ is projected to $\mathbb{R}^3$ as explained in Algorithm 2 by fixing the projection hyperplane at the position $\boldsymbol{e}_5 = (0, 0, 0, 0, 0) \in \mathbb{R}^5$ (resp. $\boldsymbol{e}_4 = (0, 0, 0, 10) \in \mathbb{R}^4$) with respect to the origin. Appendix A.3 describes how the orientation of $\mathcal{C}$ around the origin in $\mathbb{R}^n$ is described by a **general 5 by 5 rotation matrix** $\boldsymbol{R} = (\boldsymbol{R}_{i,j}) \in SO(5)$ with entries $(-1 \le \boldsymbol{R}_{i,j} \le 1)$. Therefore, at any time-step $t$ the projection of $\mathcal{C}$ is parameterized by a state

$$s_t = (d_5, d_4, \boldsymbol{R}_{1,1}, \boldsymbol{R}_{1,2}, ..., \boldsymbol{R}_{5,4}, \boldsymbol{R}_{5,5}) \in \mathbb{R}^{27}, \tag{4}$$

where the first two entries give the agent information about the camera distances and the rest are the entries of the general rotation matrix $\boldsymbol{R} = (\boldsymbol{R}_{i,j})$ (see Appendix A.3).

### 4.2 Action Space

The action space $\mathcal{A}$ describes how the agent can interact with its environment $\mathcal{E}$. The agent receives a state $s_t \in \mathcal{S}$ from $\mathcal{E}$ and selects one of the possible actions $a \in \mathcal{A}$ which then takes it to a new state $s_{t+1}$. This action space used here consists on the discrete set $\mathcal{A} := \{0, 1, ..., 17\}$; each number executes either a **camera modification** or a **5-dimensional elemental rotation**. An elemental rotation is a rotation around one of the 10 rotation planes in $\mathbb{R}^5$; and they are explained in detail in Appendix A.3. Each action $a \in \mathcal{A}$ is then identified with a vector as follows:

$$ActVec = \{0 : \delta\boldsymbol{e}^{(1)}, 1 : -\delta\boldsymbol{e}^{(1)}, 2 : \delta\boldsymbol{e}^{(2)}, 3 : -\delta\boldsymbol{e}^{(2)}, 4 : \epsilon\boldsymbol{e}^{(5)}, 5 : -\epsilon\boldsymbol{e}^{(5)}, \cdots, 16 : \epsilon\boldsymbol{e}^{(12)}, 17 : -\epsilon\boldsymbol{e}^{(12)}\}, \tag{5}$$

where $\boldsymbol{e}^{(i)}, (1 \le i \le 12, i \ne 3, 4, 7)$ are the canonical basis vectors in $\mathbb{R}^{12}$, and $\delta, \epsilon > 0 \in \mathbb{R}$ are small positive real numbers. The actions corresponding to $i = 3, 4, 7$ are discarded, because they correspond to rotations in planes $X_{0,1}, X_{0,2}, X_{1,2}$ (equivalently $Z, Y, X$) which don't yield any change on $FaceInt_{s_t}$ or $Overlap_{s_t}$.

Actions $a \in \{0, 1\}$ (resp. $a \in \{2, 3\}$) modify the distance $d_5$ (resp. $d_4$) of the 5-d (resp. 4-d) camera by a small distance $\delta > 0 \in \mathbb{R}$. To modify the 5-d (resp. 4-d) camera position we add $\pm\delta$ to the last coordinate of the camera vector, that is $\boldsymbol{c}_5[4] \leftarrow \boldsymbol{c}_5[4] + ActVec[a][0]$ (resp. $\boldsymbol{c}_4[3] \leftarrow \boldsymbol{c}_4[3] + ActVec[a][1]$).

Actions $a \in \{4, ..., 17\}$ apply a small rotation-step $\pm\epsilon > 0 \in \mathbb{R}$ in one of the planes $X_{i,j} \subset \mathbb{R}^5$. The rotation-step is taken as $\epsilon = \alpha$ degrees if at the current state $s_t$ it holds that $FaceInt_{s_t} > Exp$ and $\epsilon = 1$ degree if at the current state $s_t$ it holds that $FaceInt_{s_t} \le Exp$. Section 5 shows training plots for each surface with $\alpha = 1, 2, 5$ each, which allow to determine the best rotation-step size for each of the surfaces we study here. The change from $\epsilon = \alpha$ to $\epsilon = 1$ is intended to explore the environment in a wider extent but switching to small steps when the agent is close to a solution. To rotate $\mathcal{C}$ by an action $a \in \{4, ..., 17\}$ the entries describing the elemental rotation $ActVec[a][2, :] \in \mathbb{R}^{10}$ are selected, and the corresponding elemental rotation matrix $\boldsymbol{S} = RotMat(ActVec[a][2, :])$ is calculated as in Algorithm 1. Then $\boldsymbol{S}$ is multiplied from the left of the previous rotation matrix $\boldsymbol{R}$, and $\boldsymbol{R} \leftarrow \boldsymbol{S} \cdot \boldsymbol{R}$ is assigned. The new entries $(\boldsymbol{R}_{i,j})$ are passed to the next state $s_{t+1} \in \mathcal{S}$ as in Equation 4.

### 4.3 Reward Functions

Sparse rewards are received by the agent after achieving a polyhedral realization while dense rewards are received at each step for approaching it. In this work a combination of both is used and the reward function is the sum of the following.

Reward 5 prevents the agent from taking two consecutive inverse actions $a_t, a_{t+1}$ that yield no change in the state; for example rotating $+\epsilon$ after rotating $-\epsilon$ on the same plane $X_{i,j}$. If this is the case the agent receives a reward $r_1 = -1$.

The camera parameters $d_4$ and $d_5$ can range in the closed interval $[-15, -2] \in \mathbb{R}$. Similarly any rotation matrix $\boldsymbol{R} = (\boldsymbol{R}_{i,j}) \in SO(5)$ has entries within the interval $[-1, 1]$ (see Section A.3). The **observation**

**space** is the box $ObsSpace := [-15, -2] \times [-15, -2] \times [-1, 1]^{25} \subset \mathbb{R}^{27}$. The second reward function has to do with it, whenever the agent chooses $a_t$ such that the next state $s_{t+1} \notin ObsSpace$, then it receives a reward $r_2 = -1$ and no reward for staying inside the bounds (see Algorithm 6).

At each step, the agent attempts to reduce $FaceInt_{s_t}$ (see Algorithm 14) with respect to $FaceInt_{s_{t-1}}$. The reward in Algorithm 7 is given to the agent if $FaceInt_{s_t} > Exp$ for $Exact = False$ (resp. $FaceInt_{s_t} \neq Exp$ for $Exact = True$). If $FaceInt_{s_t} \leq Exp$ for $Exact = False$ (resp. $FaceInt_{s_t} = Exp$ for $Exact = True$) the agent does not get this reward. This reward is intended to make the agent to transition into states at which $FaceInt_{s_t}$ is closer to $Exp$ by monotonically decreasing $FaceInt_{s_t}$; although in some cases $FaceInt_{s_t}$ needs to increase in order to reach better minima like in the first row in Figure 46.

For a cubical surface $\mathcal{C}$ the task has two types of solutions depending on the *Exact* parameter. If $Exact = False$, the agent's task is to set $Overlap = 0$ (see Algorithm 13) if $FaceInt_{s_t} \leq Exp$, meaning that a realization with at most $Exp$ face intersections and without edge overlaps has been found. If $Exact = True$ it must set $Overlap = 0$ if $FaceInt_{s_t} = Exp$ (see Algorithm 8), meaning that a realization with exactly $Exp$ face intersections and without edge overlaps has been found. In both cases the agent receives a reward $r_4 = 10$. Note that the agent will tend to find solutions (terminal states $s_T$) requiring less steps because the penalization given by the future discount reward $\gamma^{t'-t}$ will be less penalized in $R_t = \sum_{t'=t}^{T} \gamma^{t'-t} r_t$.

### 4.4 Reinforcement Learning Formulation

Figure 37 shows the RL formulation of the face-intersection minimization task, showing the different flows for a $\epsilon$-degree rotation $\boldsymbol{R}_{ij}(\epsilon)$ in $\mathbb{R}^5$ or a 5-d (resp. 4-d) camera modification $\boldsymbol{c}_5[4] \leftarrow \boldsymbol{c}_5[4] \pm \delta$ (resp. $\boldsymbol{c}_4[3] \leftarrow \boldsymbol{c}_4[3] \pm \delta$). It represents a complete episode in which for a cubical surface $\mathcal{C}$ on the left, $FaceInt_{s_0} = 48$ is sequentially minimized to $FaceInt_{s_T} = 16$ with the expected face-intersection $Exp = 16$. To keep the flow diagram compact, the conditions depending on the *Exact* parameter (represented by rhomboids in gradient) can take two different criterions. A detailed pseudo-code of the environment $\mathcal{E}$ can be consulted in in Section A.7, Algorithms 9, 10 and 11.

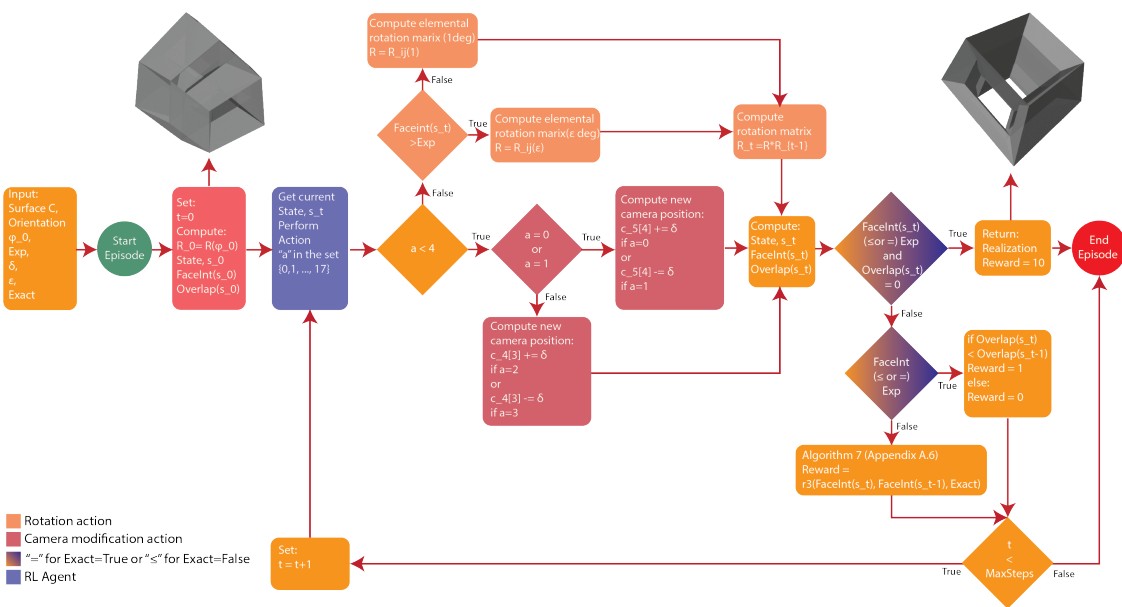

**Figure 37:** Episode flow diagram

### 4.5 The Agent

PPO (Clip) samples a size $N > 0$ batch of $s_t, a_t, \pi(a_t|s_t), r_t$ by following an initial policy $\pi_{old}$. If $Done = True$, the future discounted reward $R_t$ is calculated as in Section 4 and the episode length is saved. The

$R_t$ estimate the action-value functions $Q^{\pi_{\theta_{old}}}(s, a)$ used to calculate the advantages $\hat{A}_t$ in Equation 2. In practice, a $MinibatchSize > 0$ of elements is sampled form the memory of size $N$. According to Keskar et al. (2016) a larger $MinibatchSize$ tends to find sharper minima (leading to poor generalization), while small batch sizes tend to find flat minima (allowing better generalization); here a $MinibatchSize = 32$ is used to achieve enough generalization. Algorithm 1 in Schulman et al. (2017) shows the PPO (clip) algorithm workflow. The models were trained using Stable-Baselines 3, an implementation by Raffin et al. (2021).

## 5 Experiments

Recall from Section 4 that the rotation-step $\epsilon \in \mathbb{R}^+$ is specified by the user. A small $\epsilon$ will not be the best way to explore the entire configuration space but will be efficient to explore a particular configuration locally. On the other hand, a large $\epsilon$ will be better to explore the whole configuration space but can miss good configurations between steps. Each of the eight minimal cubical surfaces here presented is trained for 204800 steps with different rotation-step sizes, namely $\epsilon = 1$ (red), $\epsilon = 2$ (blue) and $\epsilon = 5$ (orange) degrees to find the best $\epsilon$ parameter for each surface. The number of steps sampled on the rollout function is set to $Updates = 2048$ from where a $MiniBatchSize = 32$ is sampled. During training, the initial state $s_0$ is fixed for each cubical surface and specified in Table 2, along with the initial and final $FaceInt_{s_t}$ and $Overlap_{s_t}$ values and whether the final realization is minimal in terms of $FaceInt_{s_T}$. The expected number of face intersections $Exp \geq 0$ is specified for each surface depending on the minimum $FaceInt_{s_t}$ observed in previous runs; if during training a $FaceInt_{s_t} < Exp$ is detected, then the training is repeated with the smallest $Exp$ found. The remaining training parameters are taken as: $\delta = .5$, $Exact = False$, $w = .05$, $MaxSteps = 100$, and $\gamma = .99$.

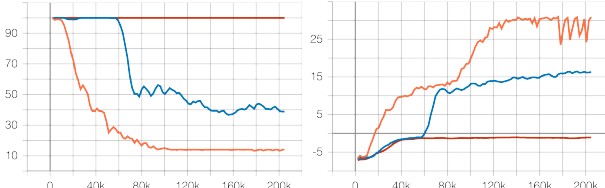

**Figure 38:** Genus-1 torus with $Exp = 0$. Left: Episode length mean. Right: Episode reward mean.

**Figure 39:** Genus-2 torus with $Exp = 0$. Left: Episode length mean. Right: Episode reward mean.

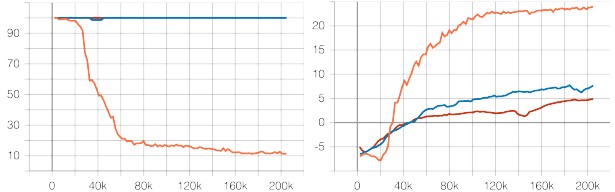

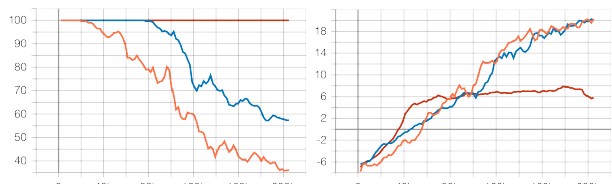

**Figure 40:** Genus-3 torus with $Exp = 9$. Left: Episode length mean. Right: Episode reward mean.

**Figure 41:** Genus-4 torus with $Exp = 12$. Left: Episode length mean. Right: Episode reward mean.

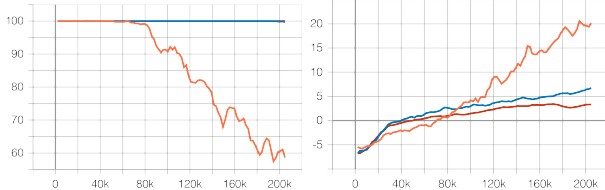

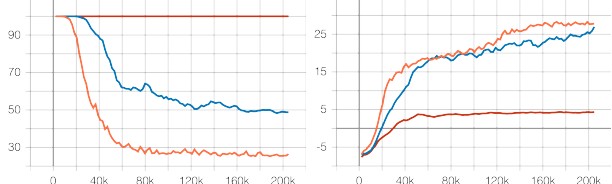

**Figure 42:** Genus-5 torus with $Exp = 16$. Left: Episode length mean. Right: Episode reward mean.

**Figure 43:** Projective Plane with $Exp = 3$. Left: Episode length mean. Right: Episode reward mean.

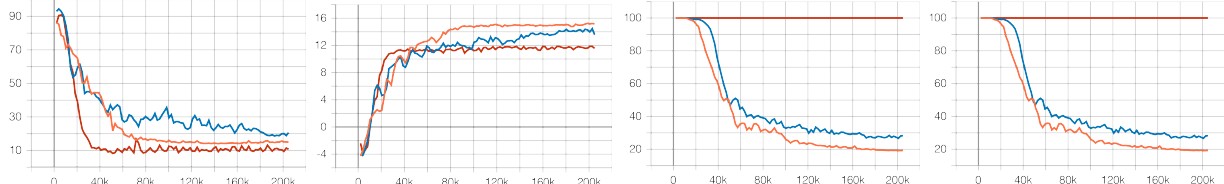

**Figure 44:** Klein Bottle with $Exp = 3$. Left: Episode length mean. Right: Episode reward mean.

**Figure 45:** K-3 surface with $Exp = 6$. Left: Episode length mean. Right: Episode reward mean.

### 5.1 An Optimization Sequence

Figure 46 shows the face intersection minimization sequence giving special emphasis on the intersection-lines for the genus-2 cubical surface in Figure 48 with $\epsilon = 5$. In this case, $FaceInt_{s_t}$ is increased only once in Figure 46c while in all other steps $FaceInt_{s_{t+1}} \leq FaceInt_{s_t}$ holds. The edge overlap minimization sequence is not presented since it is hard to appreciate it in a 2-dimensional plot. Each of the 25 steps the agent performs is a frame in an animation sequence; this allows capturing the movements induced by the 5-d rotations and perspective projections that transform $s_0$ into $s_T$ in a realistic way. Creating this animation only with the initial and final immersions (without any intermediate ones) would only yield a linear vertex displacement from their initial to the final locations, which would not correspond to the action of a 5-dimensional rotation.

### 5.2 Initial and Final Immersions of some Cubical Surfaces

The initial orientation of a cubical surface is determined from the list of angles $\phi_5 = (\phi_{0,1}, \phi_{0,2}, \phi_{0,3}, \phi_{0,4}, \phi_{1,2}, \phi_{1,3}, \phi_{1,4}, \phi_{2,3}, \phi_{2,4}, \phi_{3,4})$ sorted in lexicographic order (Section A.3). For each cubical surface and choice of rotation-step $\epsilon \in \{1, 2, 5\}$ the last trained model saved is tested. Having a high mean episode reward translates into finding a solution in less steps. Table 1 shows the number of steps each trained model needs to find a solution, showing better results for a rotation step of $\epsilon = 5$ degrees for almost all cubical surfaces tested. Table 2 compares the initial values $FaceInt_{s_0}$ and $Overlaps_{s_t}$ with the optimized $FaceInt_{s_T}$ and $Overlaps_{s_T}$ and whether the polyhedral realization is the minimal with respect to $FaceInt_{s_T}$ (under this problem's setup) or this is yet unknown.

| Surface | $\epsilon = 1$ | $\epsilon = 2$ | $\epsilon = 5$ |
|---|---|---|---|
| Genus-1 | $> 100$ | 35 | **13** |
| Genus-2 | $> 100$ | $> 100$ | **25** |
| Genus-3 | $> 100$ | $> 100$ | **11** |
| Genus-4 | $> 100$ | 56 | **39** |
| Genus-5 | $> 100$ | $> 100$ | **57** |
| Projective Plane | $> 100$ | 45 | **25** |
| Klein Bottle | **9** | 15 | 14 |
| K-3 | $> 100$ | 39 | **19** |

**Table 1:** Number of time-steps required to find a solution for a rotation step size $\epsilon$.

| $g/k$ | Initial Orientation ($\phi_5$) | $FaceInt_{s_0}$ | $FaceInt_{s_T}$ | $Overlap_{s_0}$ | $Overlap_{s_T}$ | Minimal | Figure |
|---|---|---|---|---|---|---|---|
| $g = 1$ | $(0, 0, \frac{\pi}{6}, \frac{\pi}{6}, 0, \frac{\pi}{6}, \frac{\pi}{6}, \frac{\pi}{6}, \frac{\pi}{6}, \frac{\pi}{6})$ | 6 | 0 | 2 | 0 | yes | 47 |
| $g = 2$ | $(0, 0, \frac{\pi}{6}, \frac{\pi}{6}, 0, \frac{\pi}{6}, \frac{\pi}{6}, \frac{\pi}{6}, \frac{\pi}{6}, \frac{\pi}{6})$ | 23 | 0 | 2 | 0 | yes | 48 |
| $g = 3$ | $(0, 0, 0, 0, 0, 0, \frac{\pi}{6}, \frac{\pi}{6}, \frac{\pi}{6}, \frac{\pi}{6})$ | 20 | 9 | 0 | 0 | unknown | 49 |
| $g = 4$ | $(0, 0, \frac{\pi}{6}, \frac{\pi}{6}, 0, \frac{\pi}{6}, \frac{\pi}{6}, \frac{\pi}{6}, \frac{\pi}{6}, \frac{\pi}{6})$ | 36 | 10 | 2 | 0 | unknown | 50 |
| $g = 5$ | $(0, 0, \frac{\pi}{6}, \frac{\pi}{6}, 0, \frac{\pi}{6}, \frac{\pi}{6}, \frac{\pi}{6}, \frac{\pi}{6}, \frac{\pi}{6})$ | 48 | 16 | 2 | 0 | unknown | 51 |
| $k = 1$ | $(0, 0, \frac{\pi}{6}, \frac{\pi}{6}, 0, \frac{\pi}{6}, \frac{\pi}{6}, \frac{\pi}{6}, \frac{\pi}{6}, \frac{\pi}{6})$ | 17 | 3 | 2 | 0 | yes | 52 |
| $k = 2$ | $(0, 0, \frac{\pi}{6}, \frac{\pi}{6}, 0, \frac{\pi}{6}, \frac{\pi}{6}, \frac{\pi}{6}, \frac{\pi}{6}, \frac{\pi}{6})$ | 9 | 3 | 2 | 0 | yes | 53 |
| $k = 3$ | $(0, 0, \frac{\pi}{6}, \frac{\pi}{6}, 0, \frac{\pi}{6}, \frac{\pi}{6}, \frac{\pi}{6}, \frac{\pi}{6}, \frac{\pi}{6})$ | 23 | 6 | 2 | 0 | unknown | 54 |

**Table 2:** Results for various orientable and non-orientable cubical surfaces.

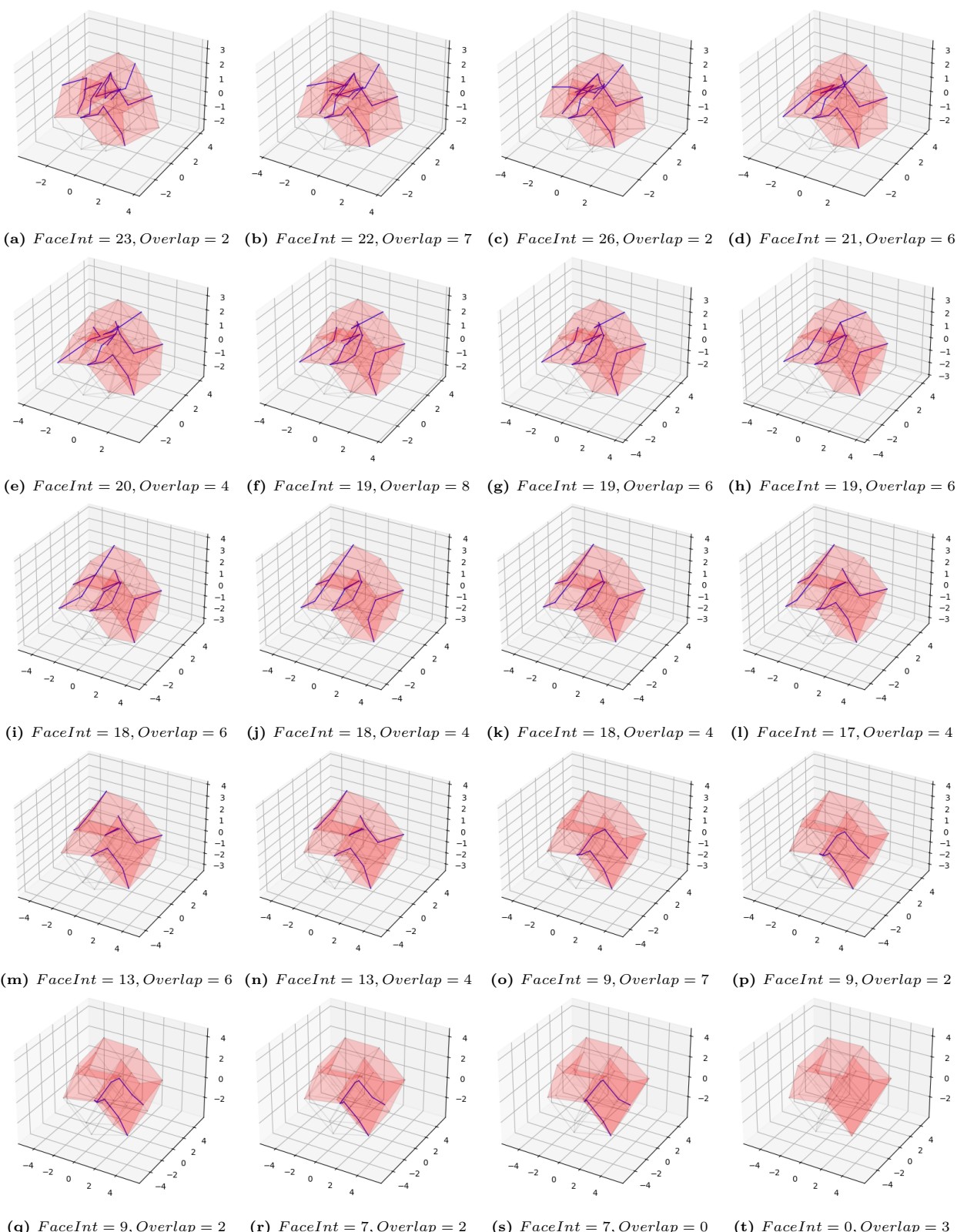

**Figure 46:** Face optimization stage for the orientable $g = 2$ cubical surface in Figure 48 with $Exp = 0$ and $\epsilon = 5$ degrees rotation-step. Faces are shown in red and face intersections in blue. The 5-cube's 1-skeleton is rendered in light gray.

## 5.3 Realizations of Orientable and Non-Orientable Cubical Surfaces

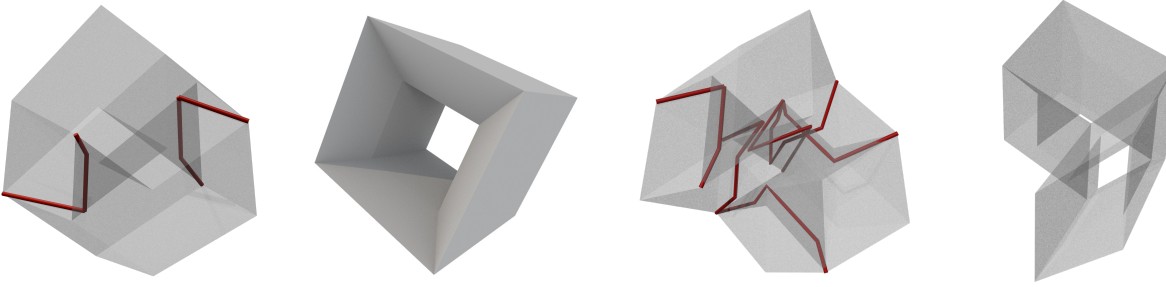

**Figure 47:** Genus-1 torus (16 faces). Left: Initial with $FaceInt = 6$. Right: Optimized with $Faceint = 0$.

**Figure 48:** Genus-2 torus (26 faces). Left: Initial with $FaceInt = 23$. Right: Optimized with $Faceint = 0$.

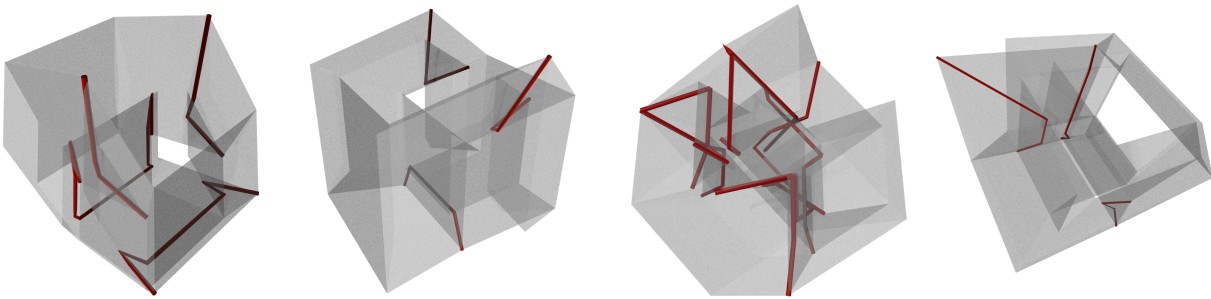

**Figure 49:** Genus-3 torus (36 faces). Left: Initial with $FaceInt = 20$. Right: Optimized with $FaceInt = 9$.

**Figure 50:** Genus-4 torus (38 faces). Left:Initial with $FaceInt = 36$. Right: Optimized with $FaceInt = 10$.

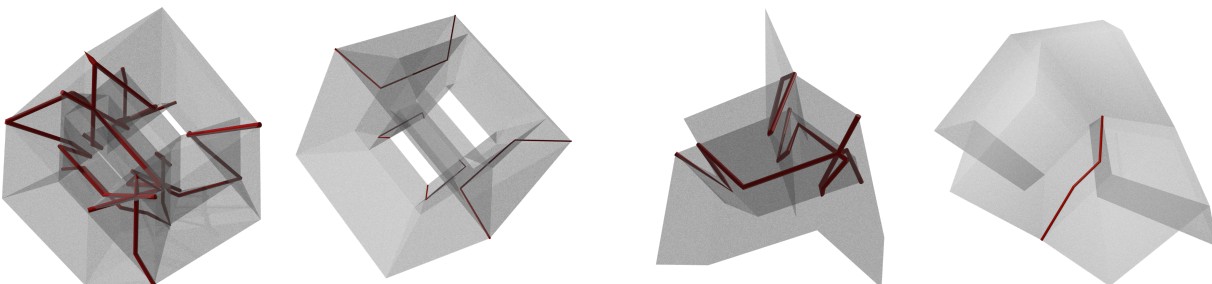

**Figure 51:** Genus-5 torus (40 faces). Left: Initial with $FaceInt = 48$. Right: Optimized with $FaceInt = 16$.

**Figure 52:** Projective Plane (20 faces). Left: Initial with $FaceInt = 17$. Right: Optimized with $FaceInt = 3$.

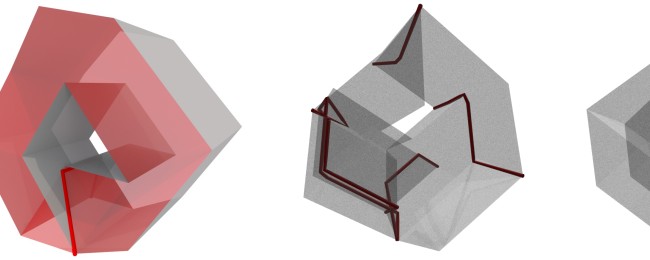
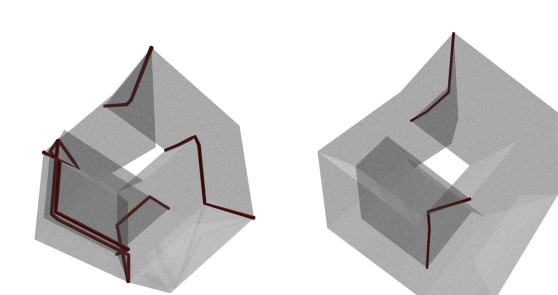

**Figure 53:** Klein Bottle (24 faces). Left: Initial with $FaceInt = 9$. Right: Optimized with $FaceInt = 3$.

**Figure 54:** K-3 surface (30 faces). Left: Initial with $FaceInt = 23$. Right: Optimized with $FaceInt = 6$.

# 6 Conclusions

This article introduces the first RL approach to find quadrilateral realizations (allowing pinch-points) of 5-dimensional orientable and non-orientable closed cubical surfaces with the smallest number of face intersections. Unlike the algorithm proposed by Hougardy et al. (2006) where the "actions" consist on moving each vertex individually in $\pm X, \pm Y$ or $\pm Z$, in this work the vertices move according to the action induced by a 5-dimensional rotation or by affecting the camera vectors $c_5 \in \mathbb{R}^5$ and $c_4 \in \mathbb{R}^4$.

Singular quadrilateral realizations of the minimal cubical projective plane and the cubical Klein Bottle with the minimum number of face intersections (3), and quadrilateral embeddings of minimal orientable cubical surfaces of genus $g = 1, 2$ are found. For orientable cubical surfaces with $g = 3, 4, 5$ and non-orientable with $k = 3$, the models here presented are candidates of singular realizations with the minimal number of face intersections. However there is no framework to prove this strictly as done here with the Klein Bottle and the Projective Plane.

For each cubical surface $\mathcal{C}$, the trained models return a sequence of steps consisting either of a camera modification or a 5-dimensional rotation, which are used to build animations. The optimal strategy has the property that it allows a face intersection increase if necessary to find a realization, still a small number of steps is required to find a realization.

# 7 Further directions

For orientable cubical surfaces with $g = 3, 4, 5$, our RL algorithm can not reduce $FaceInt_{s_t}$ below the results shown in Section 5 even though these surfaces are orientable and could be embedded allowing the unitary 5-cube to be deformed in some way. A possible solution could be a combination of the approach by Hougardy et al. (2006) in which vertices move individually in $\pm X, \pm Y$ or $\pm Z$ with RL. This would allow increasing the intersection segment functional when necessary in order to find a realization.

For all non-orientable cubical surfaces, a natural direction of interest is finding **realizations of cubical surfaces without pinch-points** as studied in Brehm & Leopold (2016) for triangulations of non-orientable surfaces with the minimal (or few) number of vertices. The modification in our algorithm would consist on finding realizations by minimizing only the face number of face intersections of faces that share a vertex which are precisely the face intersections that yield pinch-points. However, for this work the author's focus was focusing in the minimality of face intersections in general before experimenting with clearing pinch-points.

The RL algorithm is being tested with the 6-dimensional cubical surfaces from the GitHub repository by Govc (2024). Here there exist representatives of orientable surfaces with genus $g = 3$ that can be embedded in $\mathbb{R}^3$. For non-orientable cubical surfaces with $k = 1, 2$ we still observe that $FaceInt = 3$ which is still the minimum number of face intersections achievable for $Q^6$.

# 8 Supplements

If the reader wishes to explore the cubical surfaces here presented more closely, their **3-d models and animation sequences** can be consulted and downloaded from Sketchfab Estevez (2025a).

If the reader wishes to minimize a particular cubical surface or take a deeper look into the **Python implementation** visit the following GitHub repository Estevez (2025b).

Some **3-d prints** of 5-dimensional cubical surfaces can be consulted in Estévez et al. (2024).

# Acknowledgments

In loving memory of Prof. Dr. Sayan Mukherjee. This work was supported by Prof. Dr. Sayan Mukherjee's Humboldt Research Fellowship, awarded by the Alexander von Humboldt Foundation in 2023.

Thanks to Nikola Milosevic for the fruitful conversations and advices on this project and in general in RL.

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

# A Appendix

## A.1 Face Intersections

Let $\mathcal{C}$ be a cubical surface and consider projections $F_1, F_2 \subset \mathbb{R}^3$ of faces $f_1, f_2 \in \mathcal{C}$. Projected edges $e_1 \subset f_1$ and $e_2 \subset f_2$ are denoted by $E_1, E_2 \subset \mathbb{R}^3$. If $f_1, f_2 \in \mathcal{C}$ share a vertex $v$ it means that $v \in Q^n$ is one of the four vertices building $f_1$ and $f_2$. Note that if $f_1, f_2$ share three or more vertices, then $f_1 = f_2$. T*he author claims that if $F_1 \cap F_2 \neq \emptyset$, then there exist at least 3 face intersections.*

1. If $F_1$ and $F_2$ share exactly two vertices, then the following cases can occur:

   (a) If the shared vertices are common to an edge, then $F_1$ and $F_2$ are adjacent; an adjacency is not counted as a face intersection.

   (b) If the shared vertices don't share any edge, then $F_1$ and $F_2$ must intersect along their diagonal; however, this is impossible for faces $f_1, f_2 \in \mathcal{C}$ because there is a unique face connecting such two vertices.

2. If $F_1$ and $F_2$ share exactly one vertex $v$, then the following cases can occur:

   (a) $F_1 \cap F_2 = v$ with $v$ an adjacent vertex; this is not counted as an intersection.

   (b) $F_1 \cap F_2 = L$, where $L$ is a line segment.

      i. The boundaries of $F_1$ and $F_2$ intersect at some point other than $v$. This is not a valid intersection for our realization because some pair of edges would overlap as explained in Section A.2. See Figure 55.

      ii. The edges of $F_1$ and $F_2$ do not intersect at other point than $v$. Since the faces come from a projection of an $n$-dimensional cube, there can't be a third face intersecting as in Figure 58, so WLOG $F_1$ crosses the boundary of $F_2$ through edge $E_2 \subset F_2$. From (1) for a cubical surface, every edge shares exactly 2 faces, so in addition to $F_1 \cap F_2$ a second face $F_2'$ adjacent to $E_2$ intersects $F_1$ and by the observation above $F_2'$ must intersect some face $F_1'$ adjacent to $F_1$. This gives at least 3 pairs of face intersections as in Figures 56 and 57.

3. If $F_1$ and $F_2$ have no adjacent vertices, and $F_1 \cap F_2 \neq \emptyset$, then the following cases can occur:

   (a) $v \in F_1 \cap F_2$ with $v$ a non-adjacent vertex. From (2) of a cubical surface, $\mathcal{F}_v$ is a connected cyclic graph, then every vertex is shared by at least 3 faces. See Figure 59.

   (b) For a pair of edges $E_1 \subset F_1, E_2 \subset F_2$ it holds that $E_1 \cap F_2 \neq \emptyset$ and $E_2 \cap F_1 \neq \emptyset$. In addition to $F_1 \cap F_2 \neq \emptyset$ which can be a line $L$ or a point $P$; from (1) of a cubical surface every edge shares exactly 2 faces. Then $E_1 \cap F_2$ gives another intersection of $F_2$ with some other face adjacent to $E_1$ and $E_2 \cap F_1$ gives a third one of $F_1$ with some face adjacent to $E_2$. See Figure 60 and 61.

   (c) For a pair of edges $E_1 \subset F_1, E_1' \subset F_1$ it holds that $E_1 \cap F_2 \neq \emptyset$ and $E_1' \cap F_2 \neq \emptyset$. In addition to $F_1 \cap F_2 \neq \emptyset$ which can be a line $L$ or a point $P$; from (1) of a cubical surface every edge shares exactly 2 faces. Then $E_1 \cap F_2$ gives another intersection of $F_2$ with some other face adjacent to $E_1$ and $E_1' \cap F_2$ gives a third one of $F_2$ with some face adjacent to $E_1'$. See Figure 62 and 63.

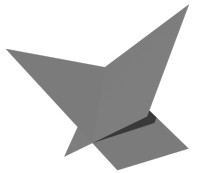 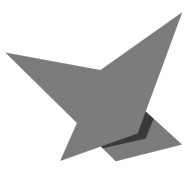 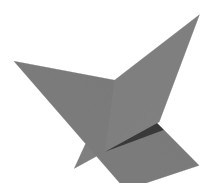 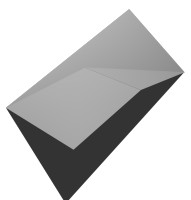

**Figure 55:** $F_1 \cap F_2 = L$ sharing a vertex; case not allowed by edge overlaps.

**Figure 56:** $F_1 \cap F_2 = L$ sharing a vertex.

**Figure 57:** $F_1 \cap F_2 = L$ sharing a vertex.

**Figure 58:** $F_1 \cap F_2 = L$ sharing a vertex; case not allowed by definition of a face.

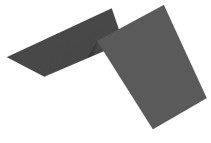 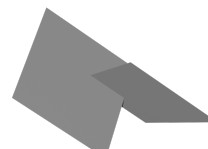 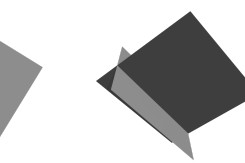 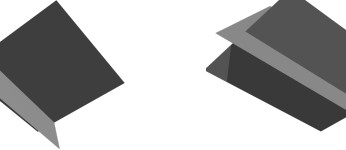

**Figure 59:** $F_1 \cap F_2 = v$ with $v$ a non-adjacent vertex.

**Figure 60:** $E_1 \cap F_2 \neq \emptyset$ and $E_2 \cap F_1 \neq \emptyset$.

**Figure 61:** $E_1 \cap F_2 \neq \emptyset$ and $E_2 \cap F_1 \neq \emptyset$.

**Figure 62:** $E_1 \cap F_2 \neq \emptyset$ and $E_1' \cap F_2 \neq \emptyset$.

**Figure 63:** $E_1 \cap F_2 \neq \emptyset$ and $E_1' \cap F_2 \neq \emptyset$.

## A.2 Edge Overlaps

Intersections between two line segments resulting from the perspective projection of two edges $e_1, e_2 \in Q_1^5$ (although more unlikely) can also occur. To model and easily visualize the projected line segments, they are assigned a "small" **edge-width** $w \geq 0 \in \mathbb{R}$. Consider points $P_1, P_2, P_3, P_4 \in \mathbb{R}^3$ and (infinite) lines $L_{1,2}$ and $L_{3,4}$ passing through $P_1, P_2$ and $P_3, P_4$ respectively. If $L_{1,2}$ and $L_{3,4}$ are co-planar and not parallel, then they intersect. However, if they are not co-planar, there exists a unique shortest line segment $L_{a,b} \subset \mathbb{R}^3$ connecting them which is perpendicular to both lines with $P_a \in L_{1,2}$ and $P_b \in L_{3,4}$. The Algorithm by Bourke (1998) described in Section A.8 computes the length of the line $|L_{a,b}| \geq 0 \in \mathbb{R}$ and whether $P_a$ (resp. $P_b$) is between points $P_1, P_2 \in L_{1,2}$ (resp. $P_3, P_4 \in L_{3,4}$). Given an edge-width $w$, two line segments $L_{1,2}, L_{3,4}$ have an **edge overlap** if $P_a \neq P_1, P_2$, $P_b \neq P_3, P_4$, $P_a \in L_{1,2}$, $P_b \in L_{3,4}$, and $|P_{a,b}| \leq 2w$. The **number of edge overlaps** at state $s_t$ denoted by $Overlap_{s_t}$ is calculated using Algorithm 12. This name is chosen to distinguish this criterion from an edge intersection; however, both have the same meaning if $w = 0$.

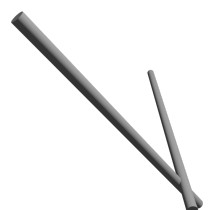

**Figure 64:** An edge overlap for edge radius $w > 0 \in \mathbb{R}$.

**Figure 65:** 3-d perspective projection of $Q_1^5$ with $Overlap = 0$ for $w = 0.5$.

## A.3  N-dimensional Rotations & Gimbal-Lock

---

**Algorithm 1:** $RotMat(\phi_n)$

---

**Data:** Dimension $n \in \mathbb{N}$ and angle list $\phi_n := (\phi_{i,j} : (i,j) \in \binom{n}{2})$.
**Result:** General rotation matrix $\boldsymbol{R} \in SO(n, \mathbb{R})$.

**1** $Comb_2(n) \leftarrow \binom{n}{2}$;
**2** $\boldsymbol{R} \leftarrow \boldsymbol{I}_n$;
**3 for** $(i,j) \in Comb_2(n)$ **do**
**4** $\quad \boldsymbol{S} \leftarrow \boldsymbol{I}_n$;
**5** $\quad \boldsymbol{S}_{i,i} \leftarrow \cos(\phi_{i,j})$;
**6** $\quad \boldsymbol{S}_{j,j} \leftarrow \cos(\phi_{i,j})$;
**7** $\quad \boldsymbol{S}_{i,j} \leftarrow -\sin(\phi_{i,j})$;
**8** $\quad \boldsymbol{S}_{j,i} \leftarrow \sin(\phi_{i,j})$;
**9** $\quad \boldsymbol{R} \leftarrow \boldsymbol{S} \cdot \boldsymbol{R}$
**10 end**

---

In $\mathbb{R}^3$, elemental rotations by angles $\phi_{0,1}, \phi_{0,2}$ and $\phi_{1,2}$ in planes $Z = 0$, $Y = 0$ and $X = 0$ correspond to elemental rotation matrices $\boldsymbol{R}_{0,1}(\phi_{0,1})$, $\boldsymbol{R}_{0,2}(\phi_{0,2}), \boldsymbol{R}_{1,2}(\phi_{1,2})$ respectively. For any angle $\phi \in [0, 2\pi)$, elemental rotation matrices satisfy the relationships $\boldsymbol{R}_{i,j}(\phi) = \boldsymbol{R}_{j,i}(-\phi)$, $\boldsymbol{R}_{i,j}^{-1}(\phi) = \boldsymbol{R}_{j,i}(\phi)$, and $\boldsymbol{R}_{i,j}(\theta)\boldsymbol{R}_{j,i}(\theta) = \boldsymbol{I}_n$ and they can be constructed as follows:

$$
\boldsymbol{R}_{i,j}(\phi) := \begin{cases} R_{k,k} = \cos(\phi) & \text{if } k = i \\ R_{l,l} = \cos(\phi) & \text{if } l = j \\ R_{k,l} = -\sin(\phi) & \text{if } k = i \text{ and } l = j \\ R_{l,k} = \sin(\phi) & \text{if } k = i \text{ and } l = j \\ R_{k,k} = 1 & \text{if } k \neq i \text{ or } k \neq j \\ R_{k,l} = 0 & \text{otherwise.} \end{cases}
$$

Consider a list of angles $\phi_n := (\phi_{i,j} : (i,j) \in \binom{n}{2})$ with the rotation angles in each plane $X_{i,j}$ ordered lexicographically, that is $(i,j) < (k,l)$ if $i < k$ or $(i = k$ and $j < l)$. A **general rotation matrix** $\boldsymbol{R} \in SO(n, \mathbb{R})$ by angles $\phi_n$ can be calculated by multiplying elemental rotation matrices $\boldsymbol{R}_{i,j}(\phi_{i,j})$ for each angle $\phi_{i,j} \in \phi_n$ on the left with respect to the order given by $\phi_n$ as shown in Equation 6. Since elemental rotation matrices generally do not commute, the order of the factors is crucial.

$$
\boldsymbol{R}(\phi_n) := \boldsymbol{R}_{n-2,n-1}(\phi_{n-2,n-1}) \cdots \boldsymbol{R}_{0,2}(\phi_{0,2})\boldsymbol{R}_{0,1}(\phi_{0,1}). \tag{6}
$$

However, when calculating general rotation matrices, some considerations must be taken into account. For example, lets consider the list of angles $\phi_3 = (\phi_{0,1}, \phi_{0,2}, \phi_{1,2}) = (\alpha, -\pi/2, \gamma)$ in $\mathbb{R}^3$. Computing the corresponding general rotation matrix results:

$$
\boldsymbol{R}(\phi_n) = \begin{bmatrix} 1 & 0 & 0 \\ 0 & \cos(\gamma) & -\sin(\gamma) \\ 0 & \sin(\gamma) & \cos(\gamma) \end{bmatrix} \begin{bmatrix} \cos(\pi/2) & 0 & \sin(\pi/2) \\ 0 & 1 & 0 \\ -\sin(\pi/2) & 0 & \cos(\pi/2) \end{bmatrix} \begin{bmatrix} \cos(\alpha) & -\sin(\alpha) & 0 \\ \sin(\alpha) & \cos(\alpha) & 0 \\ 0 & 0 & 1 \end{bmatrix} = \begin{bmatrix} 0 & 0 & 1 \\ \sin(\alpha+\gamma) & \cos(\alpha+\gamma) & 0 \\ -\cos(\alpha+\gamma) & \sin(\alpha+\gamma) & 0 \end{bmatrix}
$$

Note that by setting $\beta = -\pi/2$, affecting $\alpha$ or $\gamma$ yields the same change in the rotation matrix; moreover for any values of $\alpha$ and $\gamma$ the matrix $\boldsymbol{R}(\phi_3)$ fixes the plane $Z = 0$ and a rotation in the plane $X = 0$ is no longer possible. We have then lost a degree of freedom, and in order for $\alpha$ and $\gamma$ to have again different effects the values $\beta = \pm\pi/2$ should be avoided. This phenomenon is called Gimbal-Lock and appears in higher dimensions as well. Since in our algorithm, the rotations (see Section 4) performed by the agent step-wise are in just one plane $X_{i,j} \subset \mathbb{R}^n$ one can still use Euler angles and avoid Gimbal-Lock as explained by Shehata (2020) for cases $n = 3, 4$. The strategy is to use the action of $SO(n)$ in $\mathbb{R}^3$ when actualizing the rotation matrix at each step. By the compatibility axiom of group action, for any two rotation matrices $\boldsymbol{R}, \boldsymbol{S} \in SO(n)$ and any vector $\boldsymbol{x} \in \mathbb{R}^n$ the property $\boldsymbol{S} \cdot (\boldsymbol{R} \cdot \boldsymbol{x}) = (\boldsymbol{S} \cdot \boldsymbol{R}) \cdot \boldsymbol{x}$ holds. Instead of adding the angle $\phi_{i,j}$ to the corresponding coordinate in the list $\phi_n$ and recalculating $\boldsymbol{R}(\phi_n)$, let $\boldsymbol{R} = \boldsymbol{R}(\phi_n)$ be the previous rotation matrix and $\boldsymbol{S} = \boldsymbol{R}_{i,j}(\phi_{i,j})$ be the elemental rotation matrix for the rotation at the current step

and assigning $\boldsymbol{R} \leftarrow \boldsymbol{S} \cdot \boldsymbol{R}$. Algorithm 1 describes how to calculate a general rotation matrix $\boldsymbol{R}(\phi_n)$ given an ordered list of angles $\phi_n$ as in Equation 6. It will be used to construct the initial embedding of the surface $\mathcal{C}$ in Algorithm 9 and after each step in Algorithm 10, successfully avoiding Gimbal-Lock.

### A.4  N-dimensional Perspective Projection

---

**Algorithm 2:** $Pr_n(\boldsymbol{a}, \boldsymbol{c}, \boldsymbol{p})$

**Data:** Point to project $\boldsymbol{a} \in \mathbb{R}^n$, Camera position $\boldsymbol{c} \in \mathbb{R}^n$, Projection Plane position $\boldsymbol{p} \in \mathbb{R}^n$.
**Result:** Projected point $\boldsymbol{b} \in \mathbb{R}^{n-1}$.

**1** $\boldsymbol{d} \leftarrow \boldsymbol{a} - \boldsymbol{c}$;
**2** $\boldsymbol{M} \leftarrow \boldsymbol{I}_n$;
**3 for** $i, (0 \leq i < n - 1)$ **do**
**4** $\quad \boldsymbol{M}_{i,n} \leftarrow \boldsymbol{p}[i]/\boldsymbol{p}[n]$;
**5 end**
**6** $\boldsymbol{M}_{n,n} \leftarrow 1/\boldsymbol{p}[n]$;
**7** $\boldsymbol{f} \leftarrow \boldsymbol{M} \cdot \boldsymbol{d}$;
**8** $\boldsymbol{f} \leftarrow \boldsymbol{f}/\boldsymbol{f}[n]$;
**9** $\boldsymbol{b} \leftarrow (\boldsymbol{f}[1], ..., \boldsymbol{f}[n-1])$;

---

**Algorithm 3:** $ProjSegList(n, \boldsymbol{R}, Q_k^n, \boldsymbol{c}_n, \boldsymbol{p}_n, ..., \boldsymbol{c}_4, \boldsymbol{p}_4)$.

**Data:** Point to project $\boldsymbol{a}_i \in \mathbb{R}^i$, $n$-dimensional Rotation Matrix $\boldsymbol{R}$, $n$-dimensional Edges $Q_1^n$, Camera position $\boldsymbol{c}_i \in \mathbb{R}^i$, Orthogonal distance from origin to projection plane $p > 0 \in \mathbb{R}$.
**Result:** $list(tuple : \boldsymbol{b} \in \mathbb{R}^3)$

**1** $SegList \leftarrow list()$;
**2 for** $e \in Q_1^n$ **do**
**3** $\quad VtxList \leftarrow list()$;
**4** $\quad$ **for** $v \in e$ **do**
**5** $\quad\quad \boldsymbol{a} \leftarrow \boldsymbol{R} \cdot v$;
**6** $\quad\quad N \leftarrow n$;
**7** $\quad\quad$ **while** $N > 3$ **do**
**8** $\quad\quad\quad \boldsymbol{a} \leftarrow Pr_N(\boldsymbol{a}, \boldsymbol{c}_N, \boldsymbol{p}_N)$;
**9** $\quad\quad\quad N \leftarrow N - 1$;
**10** $\quad\quad$ **end**
**11** $\quad\quad VtxList.append(\boldsymbol{a})$;
**12** $\quad$ **end**
**13** $\quad SegList.append(VtxList)$;
**14 end**

---

If $\boldsymbol{R} \in SO(n)$ is an $n$-dimensional rotation matrix, then the edges $e \in Q_1^n$ and faces $f \in \mathcal{C}_2$ can be rotated in $\mathbb{R}^n$ applying $\boldsymbol{R}$ to each of its building vertices $v$. The rotated edges $\boldsymbol{R} \cdot e \subset \mathbb{R}^n$ or faces $\boldsymbol{R} \cdot f \subset \mathbb{R}^n$ can then be projected to $\mathbb{R}^3$ via a sequence of perspective projections $Pr_4(\cdots (Pr_{n-1}((Pr_n(v, \boldsymbol{c}_n, \boldsymbol{e}_n), \boldsymbol{c}_{n-1}, \boldsymbol{e}_{n-1}), \cdots), \boldsymbol{c}_4, \boldsymbol{e}_4) : \mathbb{R}^n \rightarrow \mathbb{R}^3$. After each mapping the points $\boldsymbol{b} \in Pr_i(\boldsymbol{a}_i, \boldsymbol{c}_i, \boldsymbol{e}_i) \subset \mathbb{R}^{i-1}$ should always map to the same side of the camera $\boldsymbol{c}_{i-1} \in \mathbb{R}^{i-1}$ in the next perspective projection; otherwise they would be inverted in the next projection. Cubical surfaces are faces on the 5-dimensional cube $Q^5$ centered at the origin with unit-length edges. It's vertices are of the form $(\pm 1/2, \pm 1/2, \pm 1/2, \pm 1/2, \pm 1/2) \in \mathbb{R}^5$ and lay on the boundary of the 5-dimensional sphere $\mathcal{S}^5$ of radius $r_5 = \sqrt{5(\pm 1/2)^2} = \sqrt{5/4} = \sqrt{5}/2$; or for the $n$-dimensional case in the boundary of the sphere $\mathcal{S}^n$ of radius $r_n = \sqrt{n}/2$. The camera position $\boldsymbol{c}_5 \in \mathbb{R}^5$ is limited to move in the line segment $(0, 0, 0, 0, d_5) \in \mathbb{R}^5$ with $d_5 \in [-15, -2]$ so the furthest a point can be projected to $\mathbb{R}^4$ occurs when the camera position is $\boldsymbol{c}_5 = (0, 0, 0, 0, -2)$. Consider the 4-dimensional sphere $\mathcal{S}^5$, one must know how far from the origin can any point $\boldsymbol{a} \in \mathcal{S}^5$ project so the range within $\boldsymbol{c}_4 \in \mathbb{R}^4$ can move can be determined. Assume the projection line $L \subset \mathbb{R}^5$ is contained in the plane $X_{1,2}$ spanned by axes $X_1, X_2 \subset \mathbb{R}^5$ (therefore $X_3, X_4, X_5 = 0$) and has

equation $X_2 - mX_1 + 2 = 0$. The plane $X_{1,2}$ intersects the 5-sphere in a circle of radius $r_5 = \sqrt{5}/2$ with equation $X_1^2 + X_2^2 - r_5^2 = 0$; we want to minimize the projection of the line $L$ parametrized by $m$ onto the line $X_2 = 0$. For any point $x = (r_5 \cos(\theta), r_5 \sin(\theta))$ in this circle with $\theta \in (-\pi, \pi)$, the line passing through $x$ and $(0, -2)$ has slope $m = (r_5 \sin(\theta) - (-2))/(r_5 \cos(\theta) - 0)(r_5 \sin(\theta) + 2)/r_5 \cos(\theta) = \tan(\theta) + 2/(r_5 \cos(\theta))$, and substituting the value of $m$ in the equation of $L$ yields $X_2 - (\tan(\theta) + 2/r_5 \cos(\theta))X_1 + 2 = 0$. This line intersects the line $X_2 = 0$ at the point $X_1 = 2/(\tan(\theta) + 2/r_5 \cos(\theta))$. As a function defined in the interval $(-\pi, \pi) \subset \mathbb{R}$ it achieves a maximum value $X_1 = 1.348$ at $\theta = -.5932 \approx -\pi/5$ radians, so the projected points $a \in \mathbb{R}^4$ would not map behind the camera $\boldsymbol{c}_4 = (0, 0, 0, -2) \in \mathbb{R}^4$; the 4-dimensional camera can be set as $\boldsymbol{c}_4 = (0, 0, 0, d_4)$ with $d_4 \in [-15, -2] \in \mathbb{R}$. After applying the sequence of perspective projections as in Section A.4, the resulting line segments (resp. plane segments) are stored in a list $SegList$ (resp. $FaceList$). This process is detailed in Algorithm 3 and in Appendix A the process of determining whether edges in $SegList$ (resp. faces in $FaceList$) intersect in $\mathbb{R}^3$ is explained. For the list of projected edges $SegList$, we assign its elements an edge radius $w > 0 \in \mathbb{R}$ and determine if the resulting cylindrical segments intersect in pairs.

## A.5 Termination Criteria

---

**Algorithm 4:** $Done(FaceInt, Exp, Overlap, Counter, Exact)$

**Data:** $(int, int, int, int, bool); (FaceInt, Exp, Overlap, Counter, Exact)$
**Result:** $bool : Done$

1   $Done \leftarrow False$;
2   **if** $Exact$;          // (1)
3   **then**
4     **if** $Overlap = 0$ *and* $FaceInt = Exp$ **then**
5       |   $Done \leftarrow True$;
6     **end**
7   **else**
8     **if** $Overlap = 0$ *and* $FaceInt \leq Exp$ **then**
9       |   $Done \leftarrow True$;
10    **end**
11 **end**
12 **if** $Counter = MaxSteps$;          // (2)
13 **then**
14   |   $Done \leftarrow True$;
15 **end**

---

There are two termination criteria. The first one has to do with finding a realization. Given a cubical surface $\mathcal{C}$ the task has two types of solutions depending on the $Exact$ parameter. If $Exact = True$, then the objective of the agent is arriving to a terminal state $s_T$ at which $Overlap_{s_T} = 0$ and $FaceInt_{s_t} = Exp$; otherwise will be arriving to a terminal state $s_T$ at which $Overlap_{s_T} = 0$ and $FaceInt_{s_t} \leq Exp$. This is formalized in Algorithm 4. The second termination criteria is an episode truncation which ends the episode once the agent exceeds a **maximum number of steps** $MaxSteps > 0 \in \mathbb{Z}$. For some surfaces we must allow the agent to explore the environment further by increasing the allowed $MaxSteps$. In this work a $MaxSteps = 100$ is tested.

## A.6 Reward Functions

---

**Algorithm 5:** $r_1(PrevAction, Action)$

---

**Data:** $(int, int) : (PrevAction, Action)$
**Result:** $r_1 \in \mathbb{R}$

**1** $r_1 \leftarrow 0$;
**2** **if** $Counter > 1$ **then**
**3**      **if** $ActVec[PrevAction] + ActVec[Action] = 0$ **then**
**4**          $r_1 \leftarrow r_1 - 1$
**5**      **end**
**6** **end**

---

**Algorithm 6:** $r_2(State, Action)$

---

**Data:** $(tuple, int) : (State, Action)$
**Result:** $r_2 \in \mathbb{R}$

**1** $r_2 \leftarrow 0$;
**2** **if** $State[0] + ActVec[Action][0] \notin ObsSpace[0]$ $and$ $State[1] + ActVec[Action][1] \notin ObsSpace[1]$ **then**
**3**      $r_2 \leftarrow r_2 - 1$;
**4** **end**

---

**Algorithm 7:** $r_3(FaceInt, PrevFaceInt, Exact)$

---

**Data:** $(int, int, bool) : (FaceInt, PrevFaceInt)$
**Result:** $r_3 \in \mathbb{R}$

**1** **if** $Exact$ **then**
**2**      **if** $FaceInt = Exp$ **then**
**3**          $r_3 \leftarrow 0$;
**4**      **else**
**5**          **if** $|FaceInt - Exp| < |PrevFaceInt - Exp|$ **then**
**6**              $r_3 \leftarrow 1$
**7**          **end**
**8**          **if** $|FaceInt - Exp| = |PrevFaceInt - Exp|$ **then**
**9**              $r_3 \leftarrow 0$
**10**          **end**
**11**          **if** $|FaceInt - Exp| > |PrevFaceInt - Exp|$ **then**
**12**              $r_3 \leftarrow -1$
**13**          **end**
**14**      **end**
**15** **else**
**16**      **if** $FaceInt > Exp$ **then**
**17**          **if** $FaceInt < PrevFaceInt$ **then**
**18**              $r_3 \leftarrow 1$
**19**          **end**
**20**          **if** $FaceInt = PrevFaceInt$ **then**
**21**              $r_3 \leftarrow 0$
**22**          **end**
**23**          **if** $FaceInt > PrevFaceInt$ **then**
**24**              $r_3 \leftarrow -1$
**25**          **end**
**26**      **else**
**27**          $r_3 \leftarrow 0$;
**28**      **end**
**29** **end**

---

---

**Algorithm 8:** $Reward_4(FaceInt, PrevFaceInt, Overlap, PrevOverlap, Exact)$

---

**Data:** $(int, int, int, int, bool) : (FaceInt, PrevFaceInt, Overlap, PrevOverlap, Exact)$

**Result:** $r_4 \in \mathbb{R}$

**1** **if** $Exact$ **then**
**2**   **if** $FaceInt = Exp$ **then**
**3**    **if** $Overlap = 0$ *or* $PrevOverlap = 0$ **then**
**4**     $r_4 \leftarrow 10$;
**5**    **else**
**6**     **if** $Overlap < PrevOverlap$ **then**
**7**      $r_4 \leftarrow 1$
**8**     **end**
**9**     **if** $Overlap = PrevOverlap$ **then**
**10**      $r_4 \leftarrow 0$
**11**     **end**
**12**     **if** $Overlap < PrevOverlap$ **then**
**13**      $r_4 \leftarrow -1$
**14**     **end**
**15**    **end**
**16**   **end**
**17** **else**
**18**   **if** $FaceInt \leq Exp$ **then**
**19**    **if** $Overlap = 0$ *or* $PrevOverlap = 0$ **then**
**20**     $Reward_4 \leftarrow 10$;
**21**    **else**
**22**     **if** $Overlap < PrevOverlap$ **then**
**23**      $r_4 \leftarrow 1$
**24**     **end**
**25**     **if** $Overlap = PrevOverlap$ **then**
**26**      $r_4 \leftarrow 0$
**27**     **end**
**28**     **if** $Overlap > PrevOverlap$ **then**
**29**      $r_4 \leftarrow -1$
**30**     **end**
**31**    **end**
**32**   **end**
**33** **end**

---

## A.7 Environment Logic

---

**Algorithm 9:** Hypercube Environment

---

**Data:** Camera distances $d_5, d_4 \in [-15, -2]$, $\phi_i \in [0, \pi/2]$; Step size $\delta, \epsilon > 0 \in \mathbb{R}$; $Exp \geq 0 \in \mathbb{Z}$;
$Exact = bool$; $w > 0 \in \mathbb{R}$; Cubical surface $\mathcal{C} = list(tuple)$.

**1** class Hypercube5{

**2**      **Constructor$(d_5, d_4, \phi_1, ..., \phi_{10}, \delta, \epsilon, Exp, Exact, \mathcal{C}_2, w)$**

**3**          $Q_1^5 \leftarrow EdgeCoordinates(Q_1^5)$;

**4**          $\boldsymbol{f} \leftarrow FaceCoordinates(\mathcal{C})$;

**5**          $ActVec \leftarrow \{0 : \delta\boldsymbol{e}^{(1)}, 1 : -\delta\boldsymbol{e}^{(1)}, 2 : \delta\boldsymbol{e}^{(2)}, 3 : -\delta\boldsymbol{e}^{(2)}, 4 : \epsilon\boldsymbol{e}^{(3)}, 5 : -\epsilon\boldsymbol{e}^{(3)}, ..., 23 : -\epsilon\boldsymbol{e}^{(12)}\}$;

**6**          $Action \leftarrow (0, ..., 17)$;

**7**          $ObsSpace \leftarrow ([-15, -2], [-15, -2], [-1, 1], ..., [-1, 1])$;

**8**          $\boldsymbol{R} \leftarrow RotMat(\phi_1, ..., \phi_{10})$;                                 `// Algorithm 1`

**9**          $State \leftarrow (d_5, d_4, \boldsymbol{R}_{1,1}, \boldsymbol{R}_{1,2}, \cdots, \boldsymbol{R}_{5,4}, \boldsymbol{R}_{5,5})$;

**10**          $\boldsymbol{c}_5 \leftarrow State[0]\boldsymbol{e}^{(5)}$;                                     `// Camera positions.`

**11**          $\boldsymbol{c}_4 \leftarrow State[1]\boldsymbol{e}^{(4)}$;

**12**          $\boldsymbol{p}_5 \leftarrow -State[0]\boldsymbol{e}^{(5)} + \boldsymbol{e}^{(5)}$ ;             `// Hyperplane positions relative to camera.`

**13**          $\boldsymbol{p}_4 \leftarrow -State[1]\boldsymbol{e}^{(4)} + 10\boldsymbol{e}^{(4)}$;

**14**          $SegList \leftarrow ProjSegList(5, \boldsymbol{R}, Q_1^5, \boldsymbol{c}_5, \boldsymbol{p}_5, \boldsymbol{c}_4, \boldsymbol{p}_4)$;            `// Algorithm 3`

**15**          $FaceList \leftarrow ProjSegList(5, \boldsymbol{R}, \boldsymbol{f}, \boldsymbol{c}_5, \boldsymbol{p}_5, \boldsymbol{c}_4, \boldsymbol{p}_4)$;             `// Algorithm 3`

**16**          $Overlaps \leftarrow EdgeIntersections(w, SegList)$;              `// Algorithm 13`

**17**          $PrevOverlaps \leftarrow Overlaps$;

**18**          $FaceInt \leftarrow FaceInt(FaceList)$;                         `// Algorithm 14`

**19**          $PrevFaceInt \leftarrow FaceInt$;

**20**          $MinFaceInt \leftarrow PrevFaceInt$;

---

**Algorithm 10:** Step function

**48** ...
**49**   $Step(Action)$ :
**50**     $Counter \leftarrow Counter + 1$;
**51**     $PrevOverlap \leftarrow Overlap$;
**52**     $PrevFaceInt \leftarrow FaceInt$;
**53**     $R_1 \leftarrow Reward_1(PrevAction, Action)$;      // Algorithm 5
**54**     $R_2 \leftarrow Reward_1(State, Action)$;      // Algorithm 6
**55**     **if** $Action \in \{0, 1\}$ *and* $c_5[4] + ActVec[Action][0] \in [-15, -2]$ **then**
**56**       $c_5 \leftarrow c_5 + ActVec[Action][0]e^{(5)}$;      // 5-dimensional camera.
**57**     **else if** $Action \in \{2, 3\}$ *and* $c_4[3] + ActVec[Action][1] \in [-15, -2]$ **then**
**58**       $c_4 \leftarrow c_4 + ActVec[Action][1]e^{(4)}$;      // 4-dimensional camera.
**59**     **else**
**60**       $S \leftarrow RotMat(ActDir[Action][2, :])$;      // Algorithm 1
**61**       $R \leftarrow S \cdot R$;
**62**     $State \leftarrow (c_5[4], c_4[3], R_{1,1}, R_{1,2}, \cdots, R_{5,4}, R_{5,5})$;
**63**     $p_5 \leftarrow -State[0]e^{(5)} + e^{(5)}$ ;      // Hyperplane positions relative to camera.
**64**     $p_4 \leftarrow -State[1]e^{(4)} + 10e^{(4)}$;
**65**     $SegList \leftarrow ProjSegList(5, R, Q_1^5, c_5, p_5, c_4, p_4)$;      // Algorithm 3
**66**     $FaceList \leftarrow ProjSegList(5, R, f, c_5, p_5, c_4, p_4)$;      // Algorithm 3
**67**     $Overlaps \leftarrow EdgeIntersections(w, SegList)$;      // Algorithm 13
**68**     $FaceInt \leftarrow FaceInt(FaceList)$;      // Algorithm 14
**69**     $MinFaceInt \leftarrow min(MinFaceInt, FaceInt)$;
**70**     $R_3 \leftarrow Reward_3(FaceInt, PrevFaceInt)$;      // Algorithm 7
**71**     $R_4 \leftarrow Reward_4(FaceInt, PrevFaceInt, Overlap, PrevOverlap)$;      // Algorithm 8
**72**     $Reward \leftarrow R_1 + R_2 + R_3 + R_4$;
**73**     $Done \leftarrow Done(FaceInt, Exp, Overlap, Counter, Exact)$;      // Algorithm 4

**Algorithm 11:** Reset function

**48** ...
**49**   $Reset(Done)$ :
**50**     $R \leftarrow RotMat(\phi_1, ..., \phi_{10})$;      // Algorithm 1
**51**     $State \leftarrow (d_5, d_4, R_{1,1}, R_{1,2}, \cdots, R_{5,4}, R_{5,5})$;
**52**     $c_5 \leftarrow State[0]e^{(5)}$;      // Camera positions.
**53**     $c_4 \leftarrow State[1]e^{(4)}$;
**54**     $p_5 \leftarrow -State[0]e^{(5)} + e^{(5)}$ ;      // Hyperplane positions relative to camera.
**55**     $p_4 \leftarrow -State[1]e^{(4)} + 10e^{(4)}$;
**56**     $SegList \leftarrow ProjSegList(5, R, Q_1^5, c_5, p_5, c_4, p_4)$;      // Algorithm 3
**57**     $FaceList \leftarrow ProjSegList(5, R, f, c_5, p_5, c_4, p_4)$;      // Algorithm 3
**58**     $Overlaps \leftarrow EdgeIntersections(w, SegList)$;      // Algorithm 13
**59**     $PrevOverlaps \leftarrow Overlaps$;
**60**     $FaceInt \leftarrow FaceInt(FaceList)$;      // Algorithm 14
**61**     $PrevFaceInt \leftarrow FaceInt$;
**62**     $MinFaceInt \leftarrow PrevFaceInt$;

## A.8 Edge Collision Detection

Here, the algorithm by Bourke (1998) to compute the shortest line between two line segments in $\mathbb{R}^3$ is presented. Two lines $L_{1,2} \subset \mathbb{R}^3$ and $L_{3,4} \subset \mathbb{R}^3$ passing through points $P_1, P_2 \in \mathbb{R}^3$ and $P_3, P_4 \in \mathbb{R}^3$ respectively generally do not intersect. If $L_{1,2}$ and $L_{3,4}$ are not parallel and they are co-planar, then they must intersect. However, if they are not co-planar, they can be connected by a unique shortest line segment $L_{a,b} \subset \mathbb{R}^3$ perpendicular to both lines with $P_a \in L_{1,2}$ and $P_b \in L_{3,4}$. The algorithm calculates the points $P_a$ and $P_b$ defining $L_{a,b}$, and determines whether the point $P_a$ (resp. $P_b$) lies between the points $P_1$ and $P_2$ (resp. $P_3$ and $P_4$) or not. First note that any point $P \in L_{1,2}$ (resp. $P' \in L_{3,4}$) between $P_1$ and $P_2$ (resp. $P_3, P_4$) is of the form

$$P = P_1 + m_a(P_2 - P_1),$$

(resp. $P' = P_3 + m_b(P_4 - P_3)$) for some real number $0 \le m_a \le 1$ (resp. $0 \le m_b \le 1$). Since the shortest line segment $L_{a,b}$ between two lines $L_{1,2}$ and $L_{3,4}$ is perpendicular to both of them, the dot product must satisfy $(P_b - P_a) \cdot (P_2 - P_1) = 0$ (resp. $(P_b - P_a) \cdot (P_4 - P_3) = 0$). By taking $P_i - P_j = (x_i - x_j, y_i - y_j, z_i - z_j)$, and setting

$$d_{ijkl}(P_i, P_j, P_k, P_l) := (P_i - P_j) \cdot (P_k - P_l) = (x_i - x_j)(x_k - x_l) + (y_i - y_j)(y_k - y_l) + (z_i - z_j)(z_k - z_l),$$

expanding the dot product we get

$$
\begin{aligned}
(P_a - P_b) \cdot (P_2 - P_1) &= (P_1 + m_a(P_2 - P_1) - P_3 - m_b(P_4 - P_3)) \cdot (P_2 - P_1) \\
&= ((P_1 - P_3) + m_a(P_2 - P_1) - m_b(P_4 - P_3)) \cdot (P_2 - P_1) \\
&= d_{1321} + m_a d_{2121} - m_b d_{4321},
\end{aligned}
$$

therefore we get the equality

$$d_{1321} + m_a d_{2121} - m_b d_{4321} = 0, \tag{7}$$

and similarly for segments $L_{3,4}$ and $L_{a,b}$

$$d_{1343} + m_a d_{4321} - m_b d_{4343} = 0. \tag{8}$$

Solving for $m_b$ in Equation 8 yields $m_b = (d_{1343} + m_a d_{4321})/d_{4343}$, and substituting the expression for $m_b$ in Equation 7 yields $m_a = (d_{1343}d_{4321} - d_{1321}d_{4343})/(d_{2121}d_{4343} - (d_{4321})^2)$. Note that $d_{4343} \ne 0$ if and only if $P_4 \ne P_3$, therefore lets analyze when can $d_{2121}d_{4343} - (d_{4321})^2 = 0$. Let $\boldsymbol{u} = P_2 - P_1$ and $\boldsymbol{v} = P_4 - P_3$, and recall that $\boldsymbol{u} \cdot \boldsymbol{v} = ||\boldsymbol{u}|| ||\boldsymbol{v}|| \cos(\theta)$, where $\theta$ is the angle between $\boldsymbol{u}$ and $\boldsymbol{v}$. Substituting in the denominator on Equation A.8, and since $\boldsymbol{u} \ne 0$ and $\boldsymbol{v} \ne 0$ we get

$$
\begin{aligned}
d_{2121}d_{4343} - (d_{4321})^2 &= (\boldsymbol{u} \cdot \boldsymbol{u})(\boldsymbol{v} \cdot \boldsymbol{v}) - (\boldsymbol{u} \cdot \boldsymbol{v})^2 \\
&= ||\boldsymbol{u}||^2 ||\boldsymbol{v}||^2 - (||\boldsymbol{u}|| ||\boldsymbol{v}|| \cos(\theta))^2 \\
&= ||\boldsymbol{u}||^2 ||\boldsymbol{v}||^2 - ||\boldsymbol{u}||^2 ||\boldsymbol{v}||^2 \cos^2(\theta) \\
&= ||\boldsymbol{u}||^2 ||\boldsymbol{v}||^2 (1 - \cos^2(\theta)) \\
&= 0,
\end{aligned}
$$

if and only if $\theta = 0$ or $\theta = \pi$, that is if and only if $\boldsymbol{u}$ and $\boldsymbol{v}$ are parallel. Algorithm 12 determines whether line segments $L_{1,2}$ and $L_{3,4}$ intersect for a given edge-with $w$.

---

**Algorithm 12:** $IntersectionLine(w, P_1, P_2, P_3, P_4)$

---

**Data:** Edge radius $w > 0 \in \mathbb{R}$ and points $P_1, P_2, P_3, P_4 \in \mathbb{R}^3$ with $(P_2 - P_1), (P_4 - P_3) \neq 0$.

**Result:** ($bool : Intersects, float : distance, tuple : P_a, tuple : P_b$)

1  $Intersects \leftarrow False$;
2  $\boldsymbol{u} = P_2 - P_1$;
3  $\boldsymbol{v} = P_4 - P_3$;
4  $d_{1343} \leftarrow d_{ijkl}(P_1, P_3, P_4, P_3)$;
5  $d_{4321} \leftarrow d_{ijkl}(P_4, P_3, P_2, P_1)$;
6  $d_{1321} \leftarrow d_{ijkl}(P_1, P_3, P_2, P_1)$;
7  $d_{4343} \leftarrow d_{ijkl}(P_4, P_3, P_4, P_3)$;
8  $b \leftarrow (d_{2121}d_{4343} - d_{4321}d_{4321})$;
9  **if** $b = 0$ ;                             `// Lines are parallel.`
10  **then**
11      $\boldsymbol{w} = P_3 - P_1$;
12      $\boldsymbol{z} = P_4 - P_1$;
13      $pr_{\boldsymbol{u}}(P_3) = \frac{\boldsymbol{u}}{||\boldsymbol{u}||} \cdot \boldsymbol{w}$;
14      $pr_{\boldsymbol{u}}(P_4) = \frac{\boldsymbol{u}}{||\boldsymbol{u}||} \cdot \boldsymbol{z}$;
15      **if** !$(||\boldsymbol{u}|| < pr_{\boldsymbol{u}}(P_3) \wedge ||\boldsymbol{u}|| < pr_{\boldsymbol{u}}(P_4) \vee pr_{\boldsymbol{u}}(P_3) < 0 \wedge pr_{\boldsymbol{u}}(P_4) < 0)$ `// L`$_{3,4}$ `projects out of L`$_{1,2}$`.`
16       **then**
17          $distance = \frac{\boldsymbol{u}}{||\boldsymbol{u}||} \times \boldsymbol{w}$;
18          **if** $distance < 2w$ **then**
19             $Intersects \leftarrow True$
20          **end**
21      **end**
22  **else**
23      $m_a \leftarrow (d_{1343}d_{4321} - d_{1321}d_{4343})/b$;
24      $m_b \leftarrow (d_{1343} + m_a d_{4321})/d_{4343}$;
25      $P_a \leftarrow P_1 + m_a \boldsymbol{u}$;
26      $P_b \leftarrow P_3 + m_b \boldsymbol{v}$;
27      **if** $(0 < m_a < 1)$ *and* $(0 < m_b < 1)$ **then**
28          $distance \leftarrow ||P_b - P_a||$;
29          **if** $distance < 2w$ **then**
30             $Intersects \leftarrow True$;
31          **end**
32      **end**
33  **end**

---

**Algorithm 13:** $EdgeIntersections(w, SegList)$

---

**Data:** Edge radius $w > 0 \in \mathbb{R}, list : SegList$.

**Result:** $int : Overlaps$

1  $EdgeCombinations \leftarrow Combinations(SegList, 2)$;
2  $Overlaps \leftarrow 0$;
3  **for** *Edge1, Edge2 in EdgeCombinations* **do**
4      $Intersection, Distance, P_a, P_b \leftarrow IntersectionLine(w, Edge1[0], Edge1[1], Edge2[0], Edge2[1])$;
5      **if** *Intersection* **then**
6          $Overlaps \leftarrow Overlaps + 1$;
7      **end**
8  **end**

### A.9   Face Collision Detection

The algorithm by Möller (1997) to calculate whether two triangles $T_1, T_2 \subset \mathbb{R}^3$ intersect is presented. If they intersect, it also returns the coordinates of their intersection line or point. Denote the vertices of $T_1$ and $T_2$ by $V_0^1, V_1^1, V_2^1$ and $V_0^2, V_1^2, V_2^2$ respectively; and the planes they lie in by $\pi_1$ and $\pi_2$ respectively. Consider vectors $\boldsymbol{u}_2 = (V_1^2 - V_0^2)$ and $\boldsymbol{v}_2 = (V_2^2 - V_0^2)$, then for any point $x \in \pi_2$ the plane equation satisfies

$$\pi_2 : N_2 \cdot x + d_2 = 0, \tag{9}$$

where $N_2 = \boldsymbol{u}_2 \times \boldsymbol{v}_2$ and $d_2 = -N_2 \cdot V_0^2$ the projection distance of the vertex $V_0^2$ over the vector $-N_2$. The signed (perpendicular) distances from the vertices $V_i^1, i = 0, 1, 2$ of the triangle $T_1$ to the plane $\pi_2$ can be computed by inserting the vertices into the equation 9, yielding

$$d_{V_i^1} = N_2 \cdot V_i^1 + d_2, i = 0, 1, 2.$$

For triangle $T_1$ and plane $\pi_2$, two possible situations can occur:

1. If $d_{V_i^1} \neq 0$ for some $i \in \{0, 1, 2\}$ then the possible sub-cases can occur:

   (a) If all $d_{V_i^1} \neq 0, i \in \{0, 1, 2\}$ have the same sign, then all vertices of $T_1$ lie on the same side of the plane $\pi_2$, so in particular $T_1$ and $T_2 \subset \pi_2$ don't intersect.

   (b) If any of the $d_{V_i^1} = 0, i \in \{0, 1, 2\}$, or has a different sign with respect to the other $d_{V_j^1}, j \neq i$, then $T_1$ and the plane $\pi_2$ intersect.

2. If all $d_{V_i^1} = 0, i = 0, 1, 2$, then $T_1$ and $T_2$ are co-planar.

Suppose both pairs $(T_1, \pi_2)$ and $(T_2, \pi_1)$ are on the situation 1(b) described above, then there exist a line $L \subset \mathbb{R}^3$ in the direction of $D := N_1 \times N_2$ such that $L \cap T_1 \neq \emptyset$ and $L \cap T_2 \neq \emptyset$ with equation $L = O + tD$, where $O$ is some point on $L$ and $t \in \mathbb{R}$. Moreover, for triangle $T_1$ there must be a vertex $V_i^1$ lying on the other side of $\pi_2$ (or in $\pi_2$) with respect to the remaining vertices $V_j^1, j \neq i$ (otherwise we would have $T_1 \cap \pi_2 = \emptyset$, which we already discarded). To keep notation simple we suppose this vertex is $V_0^1$ (resp. $V_0^2$) for $T_1$ (resp. for $T_2$), and we consider the edges $E_{0,1}^1$ and $E_{0,2}^1$ of $T_1$ (resp. $E_{0,1}^2$ and $E_{0,2}^2$ of $T_2$). The goal is to compute a scalar parameter value $t_1$ for $B = E_{0,1}^1 \cap L = O + t_1 D$. First consider the projections of the vertices onto $L$, that is

$$p_{V_i^1} = D \cdot (V_i^1 - O), i = 0, 1, 2.$$

Let $K_i^1$ be the projection of $V_i^1$ onto $\pi_2$ and note that the triangles $\Delta V_0^1 B K_0^1$ and $\Delta V_1^1 B K_1^1$ are similar, therefore we get the following equation:

$$t_1 = p_{V_0^1} + (p_{V_1^1} - p_{V_0^1}) \frac{d_{V_0^1}}{d_{V_0^1} - d_{V_1^1}}. \tag{10}$$

Similar calculations are done to compute a scalar parameter $t_2$ for $E_{0,2}^1 \cap L = O + t_2 D$. Without loss of generality we can suppose $t_1 \leq t_2$ and therefore these two parameters yield a closed interval $[t_1, t_2] \subset \mathbb{R}$ describing the intersection of $T_1$ with $L$. By computing the corresponding interval for $T_2$, the intersection between $T_1$ and $T_2$ is computed by the intersection of both intervals.

On the other hand, if both pairs $(T_1, \pi_2)$ and $(T_2, \pi_1)$ are on the situation 2, start by projecting the triangles onto the axis where their area is maximized. A 2-dimensional triangle-triangle intersection is performed, that is checking if any edge of $T_1$ intersects some edge of $T_2$; if any intersection is found then $T_1$ and $T_2$ intersect. Otherwise it only remains to check if $T_i$ is totally contained in $T_j$ by checking if some point of $T_i$ lies inside the triangle $T_j$; then all the vertices of $T_i$ should lie inside $T_j$ otherwise $T_i$ and $T_j$ should have an edge to edge intersection which we had already discarded. This Algorithm has the following Python implementation NeonRice (2020), which we name here as **TriTriIntersect** and use in Algorithm 14 which counts the number of face intersections of the projected faces.

**Algorithm 14:** $FaceInt(FaceList)$

**Data:** $list : FaceList.$
**Result:** $(int : FaceInt, list : IntersectionLine)$
1 $FaceCombinations \leftarrow Combinations(FaceList, 2);$
2 $FaceInt \leftarrow 0;$
3 $VtxList \leftarrow list();$
4 $IntersectionLine \leftarrow list();$
5 **for** *Face1, Face2 in FaceCombinations* **do**
6   $FaceTriangles1 \leftarrow Combinations(Face1, 3);$      // Combinations of triangles in Face1.
7   $FaceTriangles2 \leftarrow Combinations(Face2, 3);$      // Combinations of triangles in Face2.
8   $TriangleIntersections \leftarrow 0;$
9   $Count \leftarrow 0;$              // Avoid co-planar face intersections.
10   **for** *vertex in Face1* **do**
11    **if** *vertex not in Face2* **then**
12     $Count \leftarrow Count + 1;$
13    **end**
14   **end**
15   **if** *Count = 3 and vertex not in VtxList* **then**
16    $VtxList.append(vertex);$
17   **end**
18   **for** *T1 in FaceTriangles1* **do**
19    **for** *T2 in FaceTriangles2* **do**
20     **if** *Count $\geq$ 3* **then**
21      $Res1, Res2, Intersects \leftarrow TriTriIntersect(T1[0], T1[1], T1[2], T2[0], T2[1], T2[2]);$
22      **if** *Intersects and $|Res2[0] - Res2[1]| \geq 1e - 10$* **then**
23       $TriangleIntersections \leftarrow TriangleIntersections + 1;$
24       $IntersectionLine.append([Res2[2], Res2[3]]);$
25      **end**
26     **end**
27    **end**
28   **end**
29   **if** *TriangleIntersections > 0* **then**
30    $FaceInt \leftarrow FaceInt + 1;$
31   **end**
32 **end**

## A.10 Notation of quadrilateral realizations

Figure 47 shows the initial (left) and optimized (right) immersions of the $g = 1$ cubical surface:

$$\{ \quad (0,0,0,2,2),(0,0,1,2,2),(0,0,2,0,2),(0,0,2,1,2),(0,1,0,2,2),(0,1,1,2,2),(0,1,2,0,2),(0,1,2,1,2),$$
$$(0,2,0,2,0),(0,2,0,2,1),(0,2,1,2,0),(0,2,1,2,1),(0,2,2,0,0),(0,2,2,0,1),(0,2,2,1,0),(0,2,2,1,1)\}$$

Figure 48 shows the initial (left) and optimized (right) immersions of the $g = 2$ cubical surface:

$$\{ \quad (0,0,0,2,2),(0,0,1,2,2),(0,0,2,0,2),(0,0,2,1,2),(0,1,0,2,2),(0,1,1,2,2),(0,2,0,2,0),(0,2,0,2,1),(0,2,1,2,0),(0,2,1,2,1),$$
$$(0,2,2,0,0),(0,2,2,0,1),(0,2,2,1,0),(0,2,2,1,1),(1,1,0,2,2),(1,1,1,2,2),(1,1,2,2,0),(1,1,2,2,1),(2,1,0,0,2),(2,1,0,1,2),$$
$$(2,1,1,0,2),(2,1,1,1,2),(2,1,2,0,0),(2,1,2,0,1),(2,1,2,1,0),(2,1,2,1,1)\}$$

Figure 49 shows the initial (left) and optimized (right) immersions of the $g = 3$ cubical surface:

$$\{ \quad (0,0,0,2,2),(0,0,2,0,2),(0,0,2,1,2),(0,0,2,2,0),(0,1,0,2,2),(0,1,2,0,2),(0,1,2,1,2),(0,2,0,2,1),(0,2,1,0,2),(0,2,1,1,2),$$
$$(0,2,1,2,0),(0,2,1,2,1),(1,0,0,2,2),(1,0,2,0,2),(1,0,2,1,2),(1,0,2,2,0),(1,1,0,2,2),(1,1,2,0,2),(1,1,2,1,2),(1,2,0,2,1),$$
$$(1,2,1,0,2),(1,2,1,1,2),(1,2,1,2,0),(1,2,1,2,1),(2,0,1,2,1),(2,0,2,0,1),(2,0,2,1,1),(2,1,0,2,0),(2,1,1,2,0),(2,1,1,2,1),$$
$$(2,1,2,0,0),(2,1,2,0,1),(2,1,2,1,0),(2,1,2,1,1),(2,2,0,0,1),(2,2,0,1,1)\}$$

Figure 50 shows the initial (left) and optimized (right) immersions of the $g = 4$ cubical surface:

$$\{ \quad (0,0,0,2,2),(0,0,1,2,2),(0,0,2,0,2),(0,0,2,1,2),(0,1,0,2,2),(0,1,1,2,2),(0,1,2,1,2),(0,2,0,2,0),(0,2,0,2,1),(0,2,1,2,0),$$
$$(0,2,1,2,1),(0,2,2,0,0),(0,2,2,0,1),(1,0,0,2,2),(1,0,1,2,2),(1,0,2,0,2),(1,0,2,1,2),(1,1,0,2,2),(1,1,1,2,2),(1,1,2,1,2),$$
$$(1,2,0,2,0),(1,2,0,2,1),(1,2,1,2,0),(1,2,1,2,1),(1,2,2,0,0),(1,2,2,0,1),(2,0,2,1,0),(2,0,2,1,1),(2,1,0,0,2),(2,1,1,0,2),$$
$$(2,1,2,0,0),(2,1,2,0,1),(2,1,2,1,0),(2,1,2,1,1),(2,2,0,1,0),(2,2,0,1,1),(2,2,1,1,0),(2,2,1,1,1)\}$$

Figure 51 shows the initial (left) and optimized (right) immersions of the $g = 5$ cubical surface:

$$\{ \quad (0,0,0,2,2),(0,0,1,2,2),(0,0,2,0,2),(0,0,2,1,2),(0,1,0,2,2),(0,1,1,2,2),(0,1,2,0,2),(0,1,2,1,2),(0,2,0,2,0),(0,2,0,2,1),$$
$$(0,2,1,2,0),(0,2,1,2,1),(1,0,0,2,2),(1,0,1,2,2),(1,0,2,0,2),(1,0,2,1,2),(1,1,0,2,2),(1,1,1,2,2),(1,1,2,0,2),(1,1,2,1,2),$$
$$(1,2,0,2,0),(1,2,0,2,1),(1,2,1,2,0),(1,2,1,2,1),(2,0,2,0,0),(2,0,2,0,1),(2,0,2,1,0),(2,0,2,1,1),(2,1,2,0,0),(2,1,2,0,1),$$
$$(2,1,2,1,0),(2,1,2,1,1),(2,2,0,0,0),(2,2,0,0,1),(2,2,0,1,0),(2,2,0,1,1),(2,2,1,0,0),(2,2,1,0,1),(2,2,1,1,0),(2,2,1,1,1)$$

Figure 52 shows the initial (left) and optimized (right) immersions of the $k = 1$ cubical surface:

$$\{ \quad (0,0,0,2,2),(0,0,1,2,2),(0,0,2,0,2),(0,0,2,2,0),(0,1,2,1,2),(0,2,0,1,2),(0,2,1,2,1),(0,2,2,1,0),(0,2,2,1,1),(1,0,2,2,1),$$
$$(1,2,1,1,2),(1,2,1,2,1),(2,0,0,2,1),(2,0,1,1,2),(2,0,2,0,1),(2,0,2,1,1),(2,1,1,1,2),(2,1,1,2,1),(2,2,1,0,1),(2,2,1,1,0)\}$$

Figure 53 shows the initial (left) and optimized (right) immersions of the $k = 2$ cubical surface:

$$\{ \quad (0,0,0,2,2),(0,0,2,0,2),(0,0,2,1,2),(0,0,2,2,0),(0,1,2,0,2),(0,1,2,1,2),(0,1,2,2,0),(0,1,2,2,1),$$
$$(0,2,0,2,1),(0,2,1,0,2),(0,2,1,1,2),(0,2,1,2,0),(0,2,1,2,1),(1,0,2,2,1),(1,1,0,2,2),(1,2,0,2,1),$$
$$(2,0,1,2,1),(2,0,2,0,1),(2,0,2,1,1),(2,1,0,0,2),(2,1,0,1,2),(2,1,0,2,0),(2,2,0,0,1),(2,2,0,1,1)\}.$$

Figure 54 shows the initial (left) and optimized (right) immersions of the $k = 3$ cubical surface:

$$\{ \quad (0,0,0,2,2),(0,0,2,0,2),(0,0,2,1,2),(0,0,2,2,0),(0,1,2,0,2),(0,1,2,2,0),(0,1,2,2,1),(0,2,0,2,1),(0,2,1,0,2),(0,2,1,1,2),$$
$$(0,2,1,2,0),(0,2,1,2,1),(1,0,2,1,2),(1,1,0,2,2),(1,1,2,2,1),(1,2,0,1,2),(1,2,1,1,2),(1,2,1,2,1),(1,2,2,0,1),(1,2,2,1,0),$$
$$(2,0,1,2,1),(2,0,2,0,1),(2,0,2,1,1),(2,1,0,0,2),(2,1,0,2,0),(2,1,1,1,2),(2,1,2,1,0),(2,1,2,1,1),(2,2,0,0,1),(2,2,0,1,1)\}$$

