# OpenReview forum: "Polyhedral Embeddings and Realizations of Orientable and Non-Orientable Cubical Surfaces using Reinforcement Learning"
_TMLR — Rejected by TMLR_

### Review · Reviewer_rHcf · 2025-08-28

**Summary Of Contributions:**

The paper aims to address the problem of finding a 3D immersion of a given 5D cubical surface with self-intersections minimized. Based on my understanding, this is a challenging problem in general, and the authors formulate it as an MDP and apply RL to search for solutions.

That said, it is not fully clear what the main contributions of the work are. It is not evident what specific gap in the literature is being filled. The paper should explicitly clarify whether the main contribution is. For example:
- Are they attempting to provide the minimal self-intersection number for arbitrary instances of 5D cubical surfaces?
- Are they aiming to show that RL can be used as a tool to establish existence of such immersions?
- Or are they demonstrating that a perspective-projection–based approach is feasible and effective?

The related work section is a good starting point and situates the problem within several relevant areas. However, the problem statement, its motivation, and the benefits of solving it are not explicitly articulated. It would be useful to explain why this problem is important, who the target audience is, and what practical or theoretical value such immersions may have and without this motivation, the significance of the work is difficult to assess.

In Section 2.2, the claim that the projection mechanism based on elemental rotations generates all possible rotations in $R^n$. This requires either a proof or a citation to established results. As it stands, the statement is not substantiated.

Regarding the RL formulation, the broader picture of how RL is being applied remains somewhat unclear. Only after piecing together different parts of the paper does it become apparent that RL is being used essentially as a search mechanism in a large state space of projections, with the aim of reaching a terminal state that maximized their notion of reward (roughly minimizing face-intersection). However, it is not obvious how this fits naturally into a sequential decision-making framework, which is the usual motivation for RL methods such as PPO. This connection and why this is a sequential decision making problem needs to be clarified.

**Additional Comments:**

The idea of applying RL to mathematical problems is promising and has clear potential. However, the current manuscript requires a lot more work in development of organization, problem definition, and experimental evaluation before it is ready for publication.

**Audience:**

Yes

**Audience Explanation:**

If this is indeed an important mathematical problem, then the topic has potential. With clearer writing, a stronger presentation of the motivation, and better organization of the paper, the work could be made accessible and interesting to a broader mathematical audience within the machine learning community.

**Broader Impact Concerns:**

-

**Claims And Evidence:**

No

**Claims Explanation:**

The claims of the paper are not clearly stated or evaluated. In the conclusion, the authors mention that using RL they were able to minimize the number of face-intersections to the current known minimum for each tested instance. However, no references are provided for these known minimum values, and in the experiments only a small number of instances are tested. This limited evidence does not convincingly support the broader claim that RL can generally be used to find 3D immersions with minimized face-intersections for arbitrary 5D cubical surfaces.

In the introduction, the paper states that orientable cubical surfaces can be embedded into 3D. Yet in the results, the reported embeddings still have face-intersections greater than zero. This appears to contradict the earlier claim and should be clarified. If embeddings without intersections are not achieved in practice, it would be important to explain why, and how this relates to the theoretical possibility of embeddings for orientable surfaces.

Finally, the evaluation metric of number of "overlaps” is used, but the term is not defined. A precise definition of what counts as an overlap is necessary for readers to understand and interpret the experimental results.

**Requested Changes:**

The paper would benefit from clearer organization. As it stands, the reader has to go through the entire text before fully understanding the objective being optimized. Providing a precise and accurate problem definition at the beginning would make the work much more accessible.

There are also a number of presentation issues—such as typos, undefined functions and variables, and some incomplete sentences—that can be addressed with careful proofreading. Improving these aspects would significantly enhance readability.

The experimental section is a useful start, but it is not yet sufficiently detailed to support a strong general contribution. Expanding the experiments, providing clearer explanations of the setup, and situating the results more directly within the context of related work would make the contribution much more compelling.

Overall, while the paper is difficult to follow, the underlying idea has potential. With clearer structure, careful writing, and a more developed experimental section, the work could be much stronger and more impactful.

---

### Review · Reviewer_exoP · 2025-09-26

**Summary Of Contributions:**

The paper formulates a problem for finding minimal face intersections of a quadrilateral realization for the orientable and non-orientable 5-dimensional cubical surfaces. The game process is designed as a 5-d rotation and the camera position change for the projection and the RL process is designed for that. Some **interesting** results are showed for the realization findings.

**Additional Comments:**

I think the paper is interesting and the open-source tool is very great. I hope more shocking results and better visualization to be done for attraction for the readers in both geometry/ML fields.

**Audience:**

Yes

**Audience Explanation:**

I think it's an interesting application for RL algorithm though the geometry part is diffcult to follow. The perliminary is fine and the problem is interesting. But it is not actually so attractive for the tmlr readers especially for the ML readers. That's more like a direct application with the difficult formulation.

**Claims And Evidence:**

Yes

**Claims Explanation:**

The formulate is relatively objective with the support for the serveral papers of the realization (Hougardy et al. (2006)/Brehm & Leopold (2016)). (Though I can not find the clues about the Aveni et al. (2024)). The perliminary of the paper is very comprehensive. At least, the RL design is reasonable.

**Requested Changes:**

1. I think the perliminary can be put into the appendix. For the section, I think a direct simple example for the whole process can be showed (maybe a supplementary video is better). I think the paper is interesting but not so stocking for the ML readers.
2. The minimality should be verified for the RL method. I am not sure whether RL satisfies minimality.
3. More complex/interesting shapes can be implemented/rendered if it can be done.
4. More insights about the "game" design should be provided for the readers that are not expert in geometry.
5. The rendering can take some software but not py for better shocking visualization.
6. How about the RL comparing with the possibly existing/transferring traditional methods, can we adopt RL to the checked cases
7. Maybe the pipeline diagram can be put into the supplementary/ or just refer the document of the open-source code which should include a diagram with the function name.
8. A pipeline for the formulation can be made and RL can be one block that claims which special module is designed in RL algorithm.

---

### Review · Reviewer_9ihs · 2025-10-20

**Summary Of Contributions:**

This paper proposes a Reinforcement Learning framework to find 3D polyhedral realizations of 5D cubical surfaces, with the primary objective of minimizing the number of self-intersections among the projected quadrilateral faces. The agent controls rotations and 5D/4D perspective camera distances and is trained with PPO-Clip to reduce the number of face intersections.

Strength:
- This is the first Reinforcement Learning formulation for minimizing face intersections of cubical surface realizations, which is novel.
- The definitions of the state, action, and reward spaces are clear.

Weaknesses:
- Some methodological choices should be justified. For example, in section 3.1, e4 = (0, 0, 0, 10) is described as a "scaling factor," but the rationale behind this specific setup is not explained. How does this choice affect the solution space?
- Experiments vary only the rotation step size for 204,800 training steps per surface. There is no statistical analysis across multiple seeds or initial orientations to report variance or robustness.
- There are no baselines (e.g., greedy local search, random search) to contextualize solution quality or advantage of the proposed method.
- The choice of PPO is reasonable, but how about other policy-gradient or evolutionary strategies? Some ablation studies or a discussion would enhance this paper.

**Audience:**

Yes

**Audience Explanation:**

The findings sit at the intersection of computational geometry, topology, and Reinforcement Learning.

**Claims And Evidence:**

Yes

**Claims Explanation:**

The empirical claims are supported by clear evidence. The paper provides explicit state/action/reward definitions.

**Requested Changes:**

Please see the weakness.

---

### Author Response · Authors · 2025-09-12
**About my corrections**

Dear Reviewer,

Since the observations that you pointed are more on the general structure and contributions of the manuscript, I am still busy attending them. I estimate that at the end of next week I can upload a new corrected version, is that ok?

Many thanks

---

### Comment · Reviewer_rHcf · 2025-12-23
**On the revised paper**

Hi,

Sorry for the delay and thank you for addressing the earlier concerns. However, it's quite difficult to distinguish between the revised parts and those that remain unchanged. Furthermore, the responses to the issues I raised aren't easily identifiable in the author's comments. Additionally, the experiments appear to be completely different now.

Could you provide a clearer indication of the changes (for instance, by using blue color for the changes) and offer a summary of your responses? This would help streamline the review process and ensure we're aligned. Thank you for your ongoing efforts.

Best regards

---

### Comment · Reviewer_rHcf · 2026-01-09
**Asking for clarification from authors**

Dear authors,

Are you working on the asked revision? If so, please let us know when are you submitting the new version.

Kind regards

---

### Decision · Action_Editor_mWFU · 2026-01-02

**Recommendation:** Reject

**Audience:**

Yes

**Audience Explanation:**

This falls under the umbrella of AI for mathematics, which has a growing audience in machine learning.

**Claims And Evidence:**

No

**Claims Explanation:**

This paper applies reinforcement learning (RL) to the problem of constructing polyhedral realizations of cubical surfaces (both orientable and non-orientable) embedded in R^3 via perspective projections of 5-dimensional cubes. The objective is to minimize face-face intersections in such realizations, which is considered a challenge in geometric topology and polyhedral geometry. Traditional approaches rely on local, greedy optimization of vertex positions; the novelty here is the use of reinforcement learning to explore global deformation paths through high-dimensional configuration space.

The paper frames the problem as an MDP where the reward penalizes face intersections. It represents cubical surfaces as 2-complexes embedded in the 5-cube, and defines 5D rotations and camera transformations as the action space. The paper uses PPO to learn policies that reduce intersections. This method demonstrates near-minimal intersection counts for several cases previously unresolved.

Unfortunately, most of our reviewing team are not experts in topology or polyhedral geometry, so we do not know whether these near-minimal intersection counts are global minima. There is also ambiguity in the reviewing discussion about provably minimal intersection counts and currently known or empirically found minimal intersection counts.

I expect the paper to be revised before the final submission to accurately characterize whether claims about minimality refer to provably known minimal values, currently known minimal values, or just the local steps the RL algorithm takes to reduce intersections.

Update: we have not received any updates or answers from the author regarding concerns of reviewer rHcf, so I have to recommend that the paper be rejected.

**Resubmission Of Major Revision:**

The authors may consider submitting a major revision at a later time.